# Outer membrane protein assembly mediated by BAM-SurA complexes

Katherine L. Fenn[1,7], Jim E. Horne [1,5,7], Joel A. Crossley [1], Nils Böhringer[2,3,4], Romany J. Horne [1,6], Till F. Schäberle [2,3,4], Antonio N. Calabrese [1], Sheena E. Radford [1,8] ✉ & Neil A. Ranson [1,8] ✉

The outer membrane is a formidable barrier that protects Gram-negative bacteria against environmental threats. Its integrity requires the correct folding and insertion of outer membrane proteins (OMPs) by the membrane-embedded β-barrel assembly machinery (BAM). Unfolded OMPs are delivered to BAM by the periplasmic chaperone SurA, but how SurA and BAM work together to ensure successful OMP delivery and folding remains unclear. Here, guided by AlphaFold2 models, we use disulphide bond engineering in an attempt to trap SurA in the act of OMP delivery to BAM, and solve cryoEM structures of a series of complexes. The results suggest that SurA binds BAM at its soluble POTRA-1 domain, which may trigger conformational changes in both BAM and SurA that enable transfer of the unfolded OMP to the BAM lateral gate for insertion into the outer membrane. Mutations that disrupt the interaction between BAM and SurA result in outer membrane assembly defects, supporting the key role of SurA in outer membrane biogenesis.

Gram-negative (diderm) bacteria have a complex outer membrane (OM) that is essential for cell integrity, virulence and pathogenesis[1]. The OM is densely packed with integral outer membrane proteins (OMPs) that share a β-barrel fold[2]. These OMPs are synthesised in the cytoplasm, and secreted into the periplasm via Sec-mediated translocation. Within the periplasm, they are bound by chaperones, especially Skp and SurA[3], before being delivered to the β-barrel assembly machinery (BAM). BAM is an OM-localized, multi-protein complex which folds and inserts OMPs into the OM. The importance of this pathway is underlined by the widespread conservation of the BAM complex in all bacteria that contain a canonical OM, and conservation of both BAM and SurA in proteobacteria[4,5]. The major component of the BAM complex, BamA, is itself an OMP that is essential in *E. coli*. BamA is composed of a 16-stranded β-barrel, which, in *E. coli*, is preceded N-terminally by five polypeptide transport associated (POTRA) domains that extend into the periplasm (Supplementary Fig 1a). BAM

also contains four accessory lipoproteins, BamB-E, of which only BamD is essential in *E. coli*, and deletions of the other lipoproteins result in growth defects of varying severity[6,7]. The deletion of SurA also causes a variety of growth defects in an array of pathogenic and lab strains of *E. coli* in both aerobic and anaerobic conditions[8]. While structural studies have revealed how BamA inserts OMPs into the OM, the details of how chaperones deliver unfolded OMPs to BAM to begin insertion/ assembly remain unknown. This is especially intriguing given the lack of ATP in the periplasm, which means that these activities cannot be coordinated via ATP binding/hydrolysis, as is commonly used to control and coordinate the actions of chaperones in protein folding in the cytoplasm[9].

The BAM complex is dynamic and can adopt a continuum of structures, as revealed by cryoEM and crystallography of the purified complex in detergents and nanodiscs[10-13], by molecular dynamics (MD) simulations[11,13,14] and by Electron Paramagnetic Resonance (EPR) of the

[1]Astbury Centre for Structural Molecular Biology and School of Molecular and Cellular Biology, Faculty of Biological Sciences, University of Leeds, Leeds LS2 9JT, UK. [2]Institute for Insect Biotechnology, Justus-Liebig-University Giessen, 35392 Giessen, Germany. [3]German Center for Infection Research (DZIF), Partner Site Giessen-Marburg-Langen, 35392 Giessen, Germany. [4]Branch for Bioresources, Fraunhofer Institute for Molecular Biology and Applied Ecology (IME), 35392 Giessen, Germany. [5]Present address: Department of Biochemistry, Tennis Court Road, Cambridge CB2 1GA, UK. [6]Present address: Steinmetz Building, Granta Park, Great Abington, Cambridge CB21 6DG, UK. [7]These authors contributed equally: Katherine L. Fenn, Jim E. Horne. [8]These authors jointly supervised this work: Sheena E. Radford, Neil A. Ranson. ✉e-mail: s.e.radford@leeds.ac.uk; n.a.ranson@leeds.ac.uk

protein in situ in the *E. coli* OM[15]. These conformations can be defined by the BamA 'lateral gate', which occurs at the seam in its β-barrel domain (between β-strands 1 and 16), and by the position of POTRA-5 which lies directly beneath the β-barrel[10,11,15,16] (Supplementary Fig. 1). In the 'Lateral Open' conformation, strands β1 and β16 are separated, and POTRA-5 occludes the periplasmic opening of the BamA barrel (Supplementary Fig. 1a, b). By contrast, in the 'Lateral Closed' conformation, β1 and β16 are hydrogen bonded which closes the barrel, and POTRA-5 swings outwards, exposing a periplasmic entrance to the lumen of BamA's β-barrel (Supplementary Fig. 1c, d). Our current understanding of BAM-catalysed folding is imprecise, but the first strand of the BamA barrel, β1, recognises an incoming unfolded OMP[2], via a conserved motif (the β-signal) in the C-terminal strand of the substrate OMP[17,18]. Precisely when and where the OMP is recognised, acquires β-structure, and enters the membrane, remain unclear but ultimately a completed barrel is released into the OM (Supplementary Fig. 1e, f). During OMP folding, β1 of BAM remains engaged with the substrate β-signal and BamA adopts a 'Wide-Open' conformation in which the β1 and β16 strands that form the gate are even further apart than in the Lateral Open state[19–21] (Supplementary Fig. 1e, f).

Before a substrate OMP reaches the BAM complex, it is thought to be maintained in a folding competent state within the periplasm via interaction with chaperones[22,23]. SurA interacts with BAM, both in *E. coli* and in vitro3[,24,25], and is thought to be the major chaperone responsible for OMP delivery to BAM and hence for OM biogenesis[17,26]. How unfolded OMPs transition from their SurA-bound state in the periplasm to β1 of BAM remains unknown. SurA is a flexible, ~45 kDa protein consisting of three domains: a Core domain (comprised of segments in the N- and C-terminal regions, and two peptidyl prolyl isomerase (PPIase) domains, PPIase-1 and PPIase-2 (for a map of the SurA domains see Supplementary Fig. 1g). PPIase-1 can adopt different positions relative to the Core domain. In one SurA conformation, herein termed the 'Compact state' (Supplementary Fig. 1h), PPIase-1 is packed against the Core domain, whereas in a second, PPIase-1 is disassociated from Core, termed here the 'Extended state' (Supplementary Fig. 1i)[27–30]. Cross-linking the PPIase-1 and Core domains via a disulphide bond results in OMP assembly defects, suggesting that these inter-domain dynamics are functionally important[31].

Here we set out to determine how SurA interacts with BAM, and how it delivers its OMP clients for folding into the OM. AlphaFold2[32] was used to guide the design of disulphide cross-links which were used to trap SurA interacting with BAM in vivo. We solved a series of BAM-SurA complexes, purified directly from the bacterial OM, with and without different substrate OMPs. Combined with proteomics analysis of *E. coli* in which the BAM-SurA interaction site is deleted, bacterial growth assays of BamA/SurA variants, and single-molecule Förster resonance energy transfer (smFRET) analysis of the BAM-SurA complex, our results suggest how SurA coordinates with BAM to enable vectorial OMP folding and insertion into the bacterial OM.

## Results

### β-augmentation mediates SurA-BAM binding

AlphaFold2 has been used previously to predict the interaction between BAM and SurA, and suggested that POTRA-1 of BamA and the N-terminal residues of SurA form an interface via β-augmentation[24,33]. These N-terminal residues are unstructured in crystallographic and AlphaFold2 predicted structures of SurA[24,27,33] (Supplementary Fig. 1h, i). To test this hypothesis, pairs of Cys residues were introduced at different locations across the putative BamA-SurA interface (Fig. 1a, b) and the protein pairs each co-expressed in *E. coli* (Methods). Western blots of the resulting samples using reducing and non-reducing SDS polyacrylamide gels confirmed that disulphide bonds consistent with the proposed interaction interface are formed within cells, and hence that BAM and SurA interact, at least in part, via β-augmentation in vivo (Fig. 1c).

To determine the functional importance of the BAM-SurA interaction interface at POTRA-1 in vivo, single point mutations (to Pro, which disrupts β-strands) or deletion of the putative β-strand (residues 23–28) in SurA were made (SurA residue numbering begins with the signal sequence (residues 1–20), which is removed by signal peptidase during secretion into the periplasm) and the resulting plasmids used to complement strains in which the genes encoding SurA or BamA were deleted or depleted, respectively (Methods). Defects in OM biogenesis were then assayed using vancomycin sensitivity as a measure of the barrier function of the OM[34–37]. Deletion of the predicted interacting residues from SurA (Δ23–28) causes defects in OM assembly (Supplementary Fig. 2). Notably, the defects are as severe as deletion of the entire surA gene (ΔsurA) (Fig. 1d, e). Proline substitutions which caused the most severe OM defects were SurA(D26P) and BamA(R76P), while other positions had little or no phenotype compared to the fully complemented strains (Fig. 1d–g). Substitution of these residues in each protein with Ala (SurA(D26A) and BamA(R76A)) had no effect on OM integrity, suggesting that the interaction between the proteins is mediated by their respective polypeptide backbones, consistent with a β-strand interaction (Fig. 1f, g). Collectively, these studies suggest that SurA and BamA interact in vivo via β-augmentation between the initially unstructured residues 23–28 of SurA and residues ~75–80 that form a β-strand in BamA POTRA-1. We further show that this interaction is necessary for efficient OMP biogenesis to create a stable OM.

The effect of deleting the interaction between SurA and BamA was next investigated by measuring changes in the global *E. coli* proteome. To achieve this, we used the ΔsurA strain described above, and complemented it with the plasmids encoding either wild-type SurA or SurA(Δ23–28) (Methods), allowing us to test the effects of deletion of the entire SurA molecule, or of the critical interaction interface alone. Comparison of the proteome of these three strains (ΔsurA, ΔsurA complemented wild-type *surA* and ΔsurA complemented *surA(Δ23–28)*) (Supplementary Fig. 3) revealed that in both the ΔsurA and *surA(Δ23–28)* backgrounds the levels of several OMPs in the OM is decreased (Fig. 2a). The decrease was significant in 75% of the OMPs detected, including the major OMPs, OmpC, OmpF, LptD and LamB. At a whole proteome level, the results show that deletion of just six residues in the SurA N-terminal region that binds BamA POTRA-1 causes effects on the *E. coli* 'OMPome' that are of similar severity to deletion of the entire SurA chaperone. Deletion of SurA results in a broad sigmaE response (elevated levels of DegP, Skp, RseB, RseC, BepA and BamA) (Fig. 2a, b) consistent with the well-characterised induction of sigmaE in response to unfolded proteins in the periplasm[3,38]. In contrast, deleting residues 23–28 of SurA does not evoke such a broad sigmaE response. Indeed, only DegP is significantly increased (Fig. 2b) suggesting that SurA (Δ23–28) retains some chaperone functions, but that its inability to deliver OMPs to BAM results in their increased degradation in the periplasm by DegP. These data confirm the importance of SurA in OMP biogenesis, and show that the interaction between SurA and BamA POTRA-1 is crucial for mediating this role.

### CryoEM of BAM-SurA: the wait complexes

Given the importance of the BAM-SurA interface for OM biogenesis and the fact that BAM and SurA do not form a sufficiently stable complex to allow its co-purification from cells[3,26], the BAM-SurA complex with an in vivo formed disulphide cross-linked at residues K27C (SurA) and R76C (BamA), was purified directly from the *E. coli* OM (Supplementary Fig. 4). The structure of the complex (shown schematically in Fig. 3a) was then determined by cryoEM (Methods), revealing the structure of the BAM complex with additional density corresponding to SurA at the expected location adjacent to POTRA-1 of BamA. Interestingly, two distinct conformations of BAM-bound SurA were observed. Both conformations showed the expected BamA-SurA β-augmentation interface (Fig. 3b, c, Supplementary Fig. 3h, i), but they differ in the conformation of the bound SurA molecule. One structure

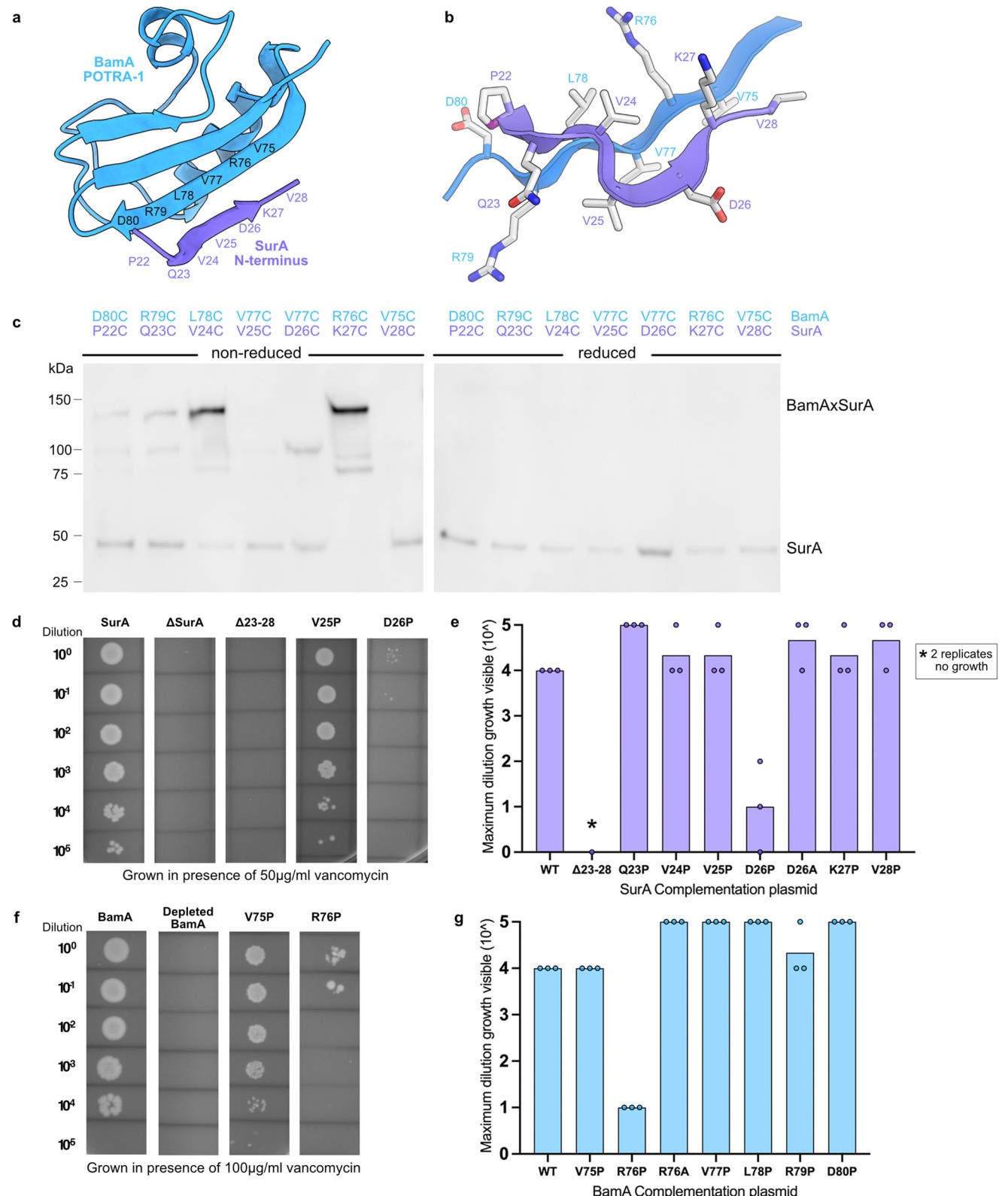

has SurA in a 'Compact' conformation (PPIase-1 bound to Core), while in the other SurA is in an 'Extended' (PPIase-1 released from Core) state (Fig. 3b, c). PPIase-2 is not resolved in either map, and PPIase-1 is also unresolved in the map of the Extended state. The interaction of SurA's Core domain with BamA is identical in both complexes, but differences are observed that propagate throughout the BAM complex. BAM is in a Lateral Open conformation in each structure, as judged by the position of POTRA-5 and the conformation at the lateral gate, and indeed in the Compact structure, a salt bridge appears to be formed between BamD and POTRA-2 (BamD-D28 and BamA-R162) which is observed in all BAM Lateral Open structures solved to date (Fig. 3d). However, release of PPIase-1 to allow SurA to adopt its Extended conformation results in POTRA-2 shifting/twisting such that it moves ~12 Å closer to turn-6 of the BamA β-barrel, breaking the BamD-D28-BamA-R162 salt bridge

**Fig. 1 | Six SurA N-terminal residues and BamA POTRA-1 interact via β-augmentation that is functionally important. a** AF2 predicted β-augmentation interface between BamA POTRA-1 (blue) and SurA residues 23–28 (purple) (numbering of SurA includes the signal sequence, which is removed during biosynthesis). **b** β-augmentation interface of edge of BamA POTRA-1 (residues 75–80) and SurA residues 22–28. **c** Western blot (using anti-strep tag antibody that recognises C-terminally Twin-Strep tagged SurA (Methods)) showing the pairs of cysteine residues introduced along the predicted β-augmentation interface in SurA and BamA POTRA-1 (in BamABCDE, named herein as BAM) that form disulphide bonds when the proteins are co-expressed in *E. coli*. Results are from *n* = 3 biological repeats. **d, e** Mutations were made in the proposed interface of SurA and BamA POTRA-1 and the resulting plasmids used to complement strains in which *surA* is deleted. Complemented strains were grown in the presence of vancomycin (50 µg/ml) to measure OM permeability defects. **d** Representative growths of bacteria (ΔsurA) complemented with plasmids expressing WT SurA, untransformed control

(ΔsurA), SurA(Δ23-28), or proline substitutions in residues 25 or 26 of SurA. (Expression levels of SurA in these strains are shown in Supplementary Fig 2a and representative growths in the absence of vancomycin are shown in Supplementary Fig 2b). **e** Maximum dilution at which growth is visible for each variant. Results are from *n* = 3 biological repeats. **f, g** Mutations were made in BamA at the proposed interface with SurA, and each variant used to complement strains in which *bamA* is depleted. Complemented strains were grown in the presence of vancomycin (100 µg/ml) to measure OM permeability defects. For SurAΔ23-28, two of the three biological repeats showed no growth even when undiluted (indicated by \*). **f** Representative growths of the WT BamA, untransformed control (depleted BamA) and two proline variants, as indicated. (Expression levels of BamA in these strains are shown in Supplementary Fig 2c and representative growths in the absence of vancomycin are shown in Supplementary Fig 2d). **g** Maximum dilution at which growth is visible for each variant. Results are from *n* = 3 biological repeats. (Source data are provided in the Source Data file for **c**, **e** and **g**).

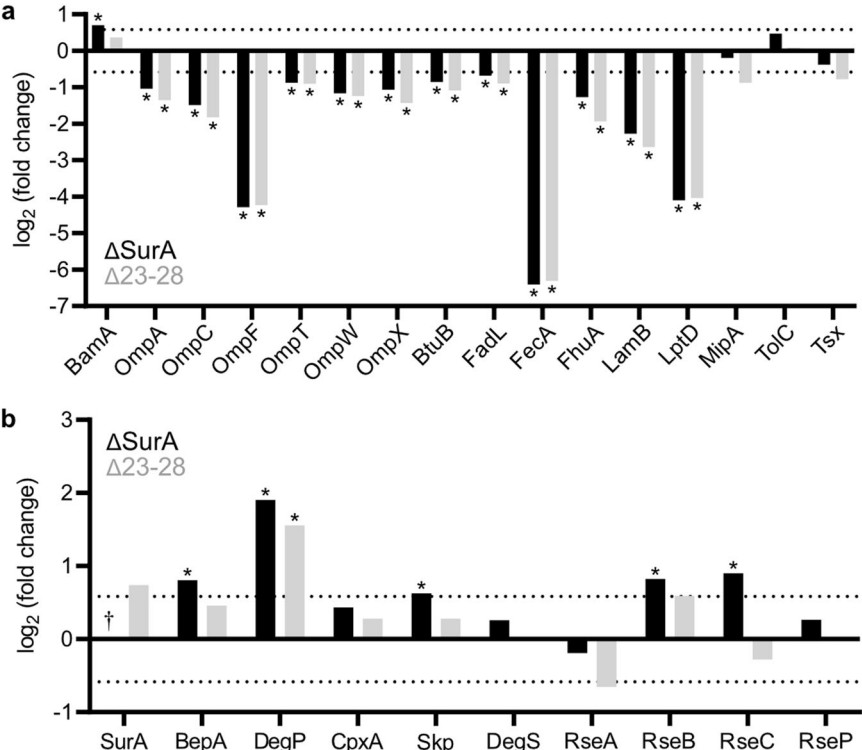

**Fig. 2 | Truncation of the six N-terminal residues of SurA is as severe as deleting SurA on the *E. coli* 'OMPome'. a** The levels of OMPs in ΔSurA (black bars) and a ΔSurA strain complemented with SurA(Δ23-28) (grey bars). **b** The levels of periplasmic chaperones and sigmaE response proteins in ΔSurA and ΔSurA strain

complemented with SurA(Δ23-28). In both **a** and **b** the values are shown relative to levels in ΔSurA complemented with wild-type SurA. \* indicates significant difference when >1.5 fold change and *p* < 0.05. (A two-sided two-sample *t*-test was used.) † indicates not detected in the ΔSurA strain.

(Fig. 3e). The Core domain interacts similarly with both POTRA-1 and POTRA-2 of BamA in the two structures, forming a discrete structural unit. However, the position of this unit relative to BamB, and the resulting interactions made between the SurA Core domain and BamB, is different, with SurA bound at blades 7 and 8 of the β-propellor of BamB in the Compact state, (Fig. 3f), but moving to blades 6 and 7 in the Extended state (Fig. 3g). This shifting around BamB results in an ~18 Å movement of POTRA-1 between the two structures. In light of these conformational changes, in the absence of a bound OMP client, we refer to these BAM-SurA structures as the 'Wait complexes'.

**The conformation of SurA is unchanged by global changes in the conformation of BAM**
The transition between the Lateral Open and Lateral Closed states of BAM is crucial for its function[16,39], and constitutes a major

rearrangement that changes the position or conformation of every component in the complex. However, how this transition is controlled remains unknown. To investigate whether the conformations of SurA bound to BAM observed above are altered when BAM adopts a Lateral Closed state, the bactericidal compound, darobactin-B (DAR-B)[40,41] was added to the POTRA-1 cross-linked SurA sample and the structure determined by cryoEM (Supplementary Fig. 5). DAR-B has been shown previously to cause complete closure of the BAM complex[15]. As expected, therefore, addition of DAR-B yielded BAM in a Lateral Closed conformation, wherein DAR-B is bound to β-strand 1 of BamA's β-barrel[15,42,43]. Identical Compact and Extended conformations of SurA bound to BAM were observed in the presence of the inhibitor, as were the differences in interaction between POTRA-2 and BamD in the SurA Extended and Compact states (Fig. 3h, i). Hence, the two SurA conformations we observe (and their relative proportions) do not change

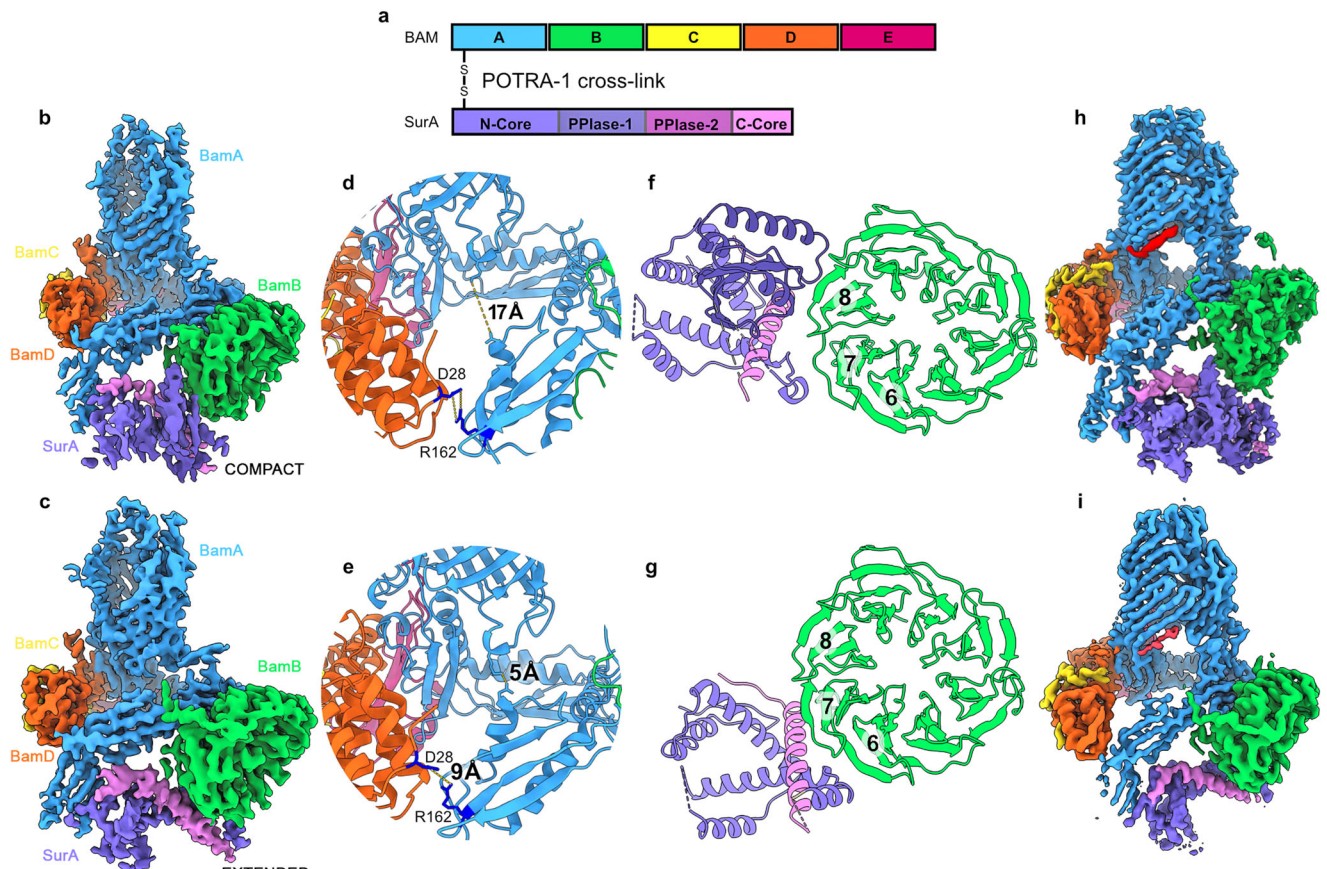

**Fig. 3 | SurA-BAM Wait Complexes. a** Schematic of BamA and SurA tethered via a disulphide bond linking residues K27C (SurA) and R76C (BamA) (not to scale). The complex was purified directly from the *E. coli* OM. The same colours are used throughout for the different proteins and their domains. CryoEM density map of **b**. Compact SurA bound to Lateral Open BAM and **c**. Extended SurA bound to Lateral Open BAM. **d** In Compact SurA bound BAM, POTRA-2 forms a salt bridge with BamD and is ~ 17 Å (measured between Cα of D107 and K726) from turn 6 of the BamA

barrel (labelled). **e** In Extended SurA bound BAM, POTRA-2 no longer forms a salt bridge with BamD and is instead twisted such that it is located ~ 5 Å from turn 6 of the BamA barrel. **f** Compact SurA bound to BAM interacts with BamB at propellers 7 and 8. **g** Extended SurA bound BAM interacts with BamB at propellers 6 and 7. **h** CryoEM density map of Compact SurA bound to Lateral Closed BAM in the presence of darobactin-B (red). **i** CryoEM density map of Extended SurA bound to Lateral Closed BAM in presence of darobactin-B (red).

when BAM is DAR-B bound, despite BAM itself undergoing a major, multi-domain rearrangement.

The addition of DAR-B to BAM traps BamA in the Lateral Closed state, decreasing flexibility of the barrel[15]. This increased rigidity is observed in our structure, and allowed the resolution of the complex to extend beyond 3Å, enabling better visualisation of SurA in its BAM-bound state. Rather than two discrete structures, BAM-SurA is instead resolvable in a continuum of states, in which the PPIase-1 domain appears to be associating and disassociating with the Core domain concurrently with the movement of POTRA-1 and POTRA-2 (Supplementary Fig. 5). Interestingly, the addition of DAR-B to this sample (2-fold molar excess) yielded ~ 25% of particles in a Lateral Open conformation, whereas no Lateral Open particles were observed in the presence of the same concentration of DAR-B when SurA is absent (Supplementary Fig. 5)[15]. This suggests that the presence of SurA may shift the equilibrium between the Lateral Open and Lateral Closed conformations of BAM towards the open state.

## BAM modulates SurA conformational states in solution

We next investigated whether binding to BAM alters the domain organisation of SurA in solution in the absence of the trapping disulphide bond. To achieve this, we used smFRET to probe inter-domain distances in the chaperone with/without BAM (SurA and BAM bind in vitro with a ~ 2.6 μM K$_d$[30]). Cys residues were introduced into SurA at position 85 (in the Core domain) and 193 (in PPIase-1) or 301 (in PPIase-

2) (Fig. 4a). These Cys residues were then labelled stochastically with donor and acceptor dyes and smFRET was used to measure the proximity ratio (PR) between the two dyes in SurA alone, or SurA bound to BAM (Methods). Consistent with previous experiments using the same dye pair[30], the results showed that SurA is predominantly (~ 80 %) in a Compact form (PR = 0.55) in the absence of BAM, with a minor population (~ 20 %), of a more Extended conformation (PR = 0.35) (Fig. 4b). The addition of BAM (75 pM labelled SurA, 7.5 μM BAM) shifts this ensemble, with the population of Extended SurA increasing (to ~ 55 %), consistent with the results observed using cryoEM and described above (Fig. 4c). By contrast, no change in the PR distribution was observed for SurA labelled on Core and PPIase-2, with or without the addition of BAM (Fig. 4,e). This suggests that PPIase-2 remains distant from Core/PPIase-1 and that its conformation is therefore not sensitive to BAM binding, at least as detected here using smFRET. In summary, these data suggest that binding of SurA to BAM shifts the conformational equilibrium of its Core and PPIase-1 domains, increasing the population of the Extended state, which in turn exposes its proposed OMP binding regions[30,44].

## SurA-OMP binding: the Arrival complex

How the presence of a substrate affects the structure of the BAM-SurA complex was next investigated by addition of the peptide, WEYIPNV, which has been shown previously to bind SurA with a 1–10 μM K$_d$[45]. Results from crystallography have shown that this peptide binds to

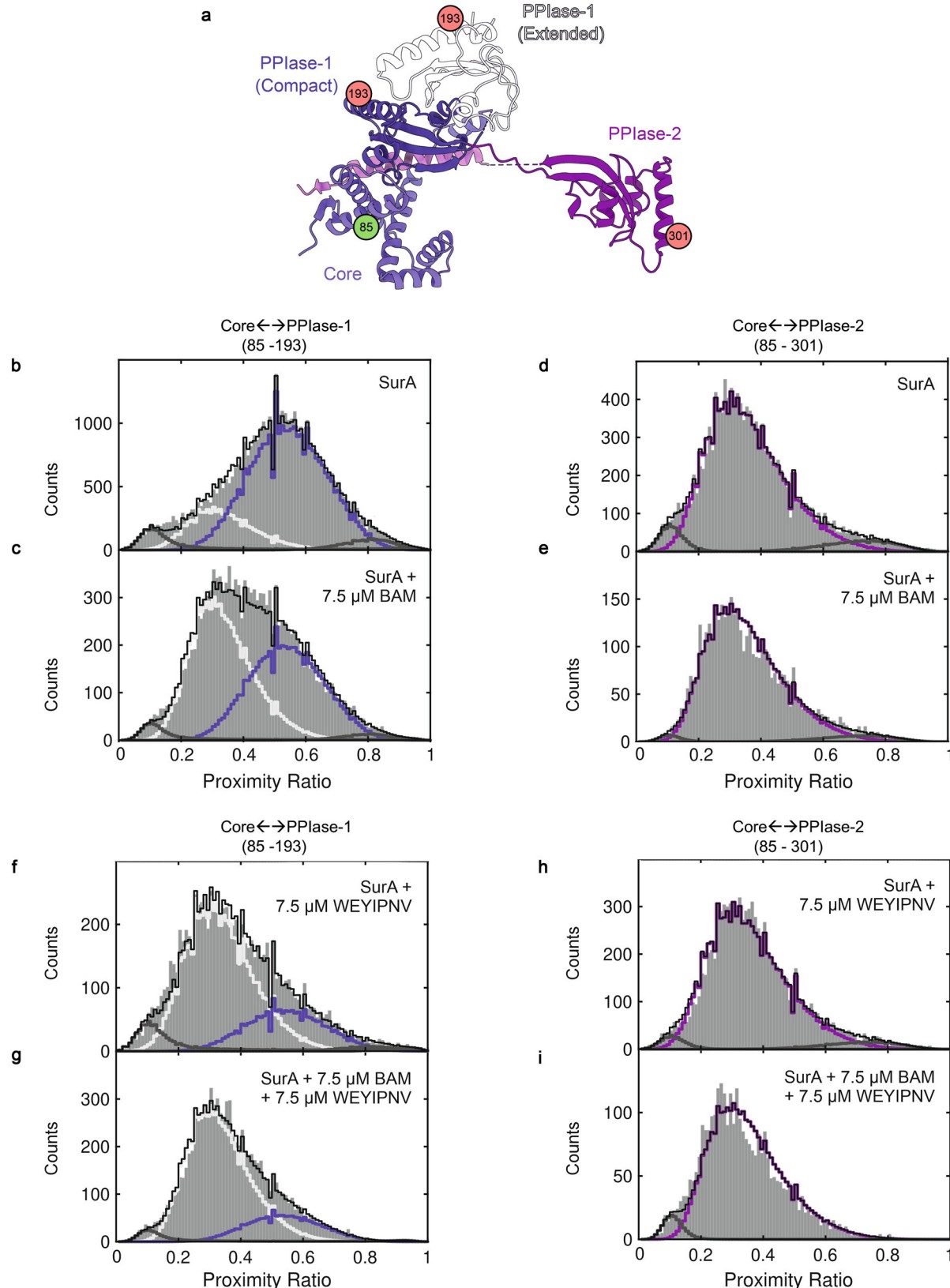

SurA PPIase-1, mimicking the binding of an OMP client and resulting in SurA adopting its Extended conformation[29,30,45,46]. It should be noted that this approach is preferable to using an intact OMP which would fold into the detergent micelle that stabilises BAM, without the addition of high concentrations of urea. We recapitulated these results here using smFRET (Fig. 4d). WEYIPNV was then added to the POTRA-1

cross-linked SurA sample and the structure determined using cryoEM. The resulting structure contained SurA in the Extended conformation, with no evidence for Compact BAM-SurA complexes that were observed in the absence of the peptide (Supplementary Fig. 6). No density for the bound peptide was observed since WEYIPNV exclusively binds PPIase-1[29] which is not resolved in this structure

**Fig. 4 | Single-molecule FRET of SurA. a** Cartoon representation of SurA, highlighting the positions of the FRET dyes in the Core (residue 85), PPIase-1 (residue 193), and PPIase-2 (residue 301) domains. For PP1ase-1 the positions in both the Extended and Compact Conformations are shown. **b–e** Proximity ratio (PR) histograms describing the distance between FRET dyes in SurA alone or in the presence of BAM. **b** SurA(Core↔PPIase-1) **c** SurA(Core↔PPIase-1) + BAM **d** SurA(Core↔PPIase-2) **e** SurA(Core↔PPIase-2) + BAM. **f–I** PR histograms describing the distance between FRET dyes in the presence of SurA plus WEYIPNV, with and without BAM. **f** SurA(Core↔PPIase-1) + WEYIPNV. **g** SurA(Core↔PPIase-1) + WEYIPNV + BAM. **h** SurA(Core↔PPIase-2) + WEYIPNV. **i** SurA(Core↔PPIase-2) + BAM. All data are globally fitted to four states for Core↔PPIase-1, or three states for Core↔PPIase-2 (Methods). The fitted states at high and low PR values shown in dark grey lines are not interpreted as structural states, owing to their low abundance (mean ˜ 4 %). Other states are shown as violet or grey lines, with the sum of all states shown as a black line. These low abundance states may arise from blinking or bleaching of the fluorescent dyes. For SurA + BAM samples, ˜ 74 % of SurA in the experiment is bound to BAM, and for SurA + WEYIPNV, ˜ 77 % of the SurA is bound to the peptide. In each case this was calculated from the known $K_d$ values for the interacting pairs (SurA+BAM = 2.6 ± 0.2 μM[24]; SurA + WEYIPNV = 2.30 ± 0.05 μM (at 20 °C)[44]. Note that in the Core↔PPIase-2 samples containing BAM, the proximity ratio histograms shift to slightly lower values, suggestive of small changes in the overall ensemble that we are unable to interpret further here.

(Supplementary Fig. 6). Consistent with this finding, addition of WEYIPNV to dye-labelled SurA for smFRET studies, pushes the SurA ensemble towards the Extended conformation (˜ 75% populated) both with and without BAM, (Fig. 4f, g). Again, no change in the PR distribution was observed for SurA labelled on Core and PPIase-2, with or without the addition of BAM in the presence of WEYINPV (Fig. 4h, i). The cryoEM and smFRET data thus show that binding of WEYIPNV to SurA PPIase-1 causes its dissociation from the Core and suggest that SurA in the Extended conformation delivers OMPs to BAM.

To interrogate the conformational state of SurA when a substrate OMP is present, one approach would be to add an unfolded OMP to the purified disulphide linked BAM-SurA complex. However, there are two problems with such a strategy. Firstly, the OMP could rapidly fold into the detergent micelle and, secondly, the affinity of SurA for a substrate OMP is in the low micromolar range, hence high concentrations (e.g. in excess of 100 μM) of unfolded OMP would be needed to ensure high occupancy on the SurA-BAM complex. Therefore, we created a chimeric SurA-OMP construct by concatenating the mature sequence of OmpX directly to the C-terminus of SurA. This increases the local concentration of the OMP, and prevents it from complete folding, effectively trapping the OMP on the BAM-SurA complex. A cryoEM dataset was collected, which contained a BAM: SurA-OmpX complex in a single conformation, with BAM in its Lateral Open conformation and SurA in its Extended state (Fig. 5b). Thus, for both free- and BAM-bound SurA, the binding of a substrate OMP (or a peptide mimic of substrate) appears to select for the extended conformation of SurA.

Additional density was present adjacent to the Core domain of SurA in the BAM: SurA-OmpX complex (Fig. 5c), consistent with a short region of the unfolded OmpX polypeptide binding at this site, although the resolution in this region of the map was not sufficient to sequence the bound peptide or determine its directionality. The SurA residues interacting with this OmpX density are consistent with previous in vitro cross-linking of a SurA-OmpX complex[30,44]. No additional density was observed at β1 of BAM, indicating that the OMP is not yet stably engaged at the site of membrane insertion. We term this structure the 'Arrival Complex', as it shows a substrate OMP bound to SurA on BAM, presumably in the initial stages of its journey to the OM, and before its β-signal engages with BamA-β1.

### The handover complex
To determine whether the SurA-OmpX hybrid is compatible with substrate engagement at BamA-β1, and thus competent to begin folding, we engineered a new Cys into β1 of BamA and a second Cys into the β-signal in the C-terminal β-strand of OmpX (BamA-S425C and OmpX-R170C). This complex of BAM and SurA-OmpX is therefore only tethered via a disulphide bond between OmpX and BamA at the site of OMP insertion to the membrane (Fig. 5d). Such a strategy has been used previously for cryoEM structure determination of BAM-substrate hybrid barrels[13,19–21]. In addition, we deleted loop 1 of OmpX to ensure we trapped OmpX folding on BAM, since deletion of this loop has been shown previously to prevent full folding of OMPs into the bilayer (Fig. 5d)[19,20]. After co-expression of these proteins, the complex was again purified directly from the *E. coli* OM (Supplementary Fig. 8a), which yielded a mixture of disulphide bonded and non-disulphide bonded complexes (Supplementary Fig. 8b). Accordingly, two classes were separated by cryoEM (Supplementary Fig. 8c–g). In one, BAM is in a Lateral Wide-open conformation, in which the BamA lateral gate forms a continuous β-sheet with residues in the C-terminal region of OmpX (Fig. 5e). Three β-strands of OmpX were resolved, corresponding to β-strands 8, 7 and 6 (Fig. 5f). Remarkably, density for OmpX was again observed in the Core domain of SurA in the same location as that seen in the Arrival complex, indicating that this structure has captured BAM in the act of folding an OMP whilst SurA is still involved in binding and chaperoning its client OMP and remains bound to BamA POTRA-1. We term this the 'Handover Complex', reflecting a handover of the substrate from SurA to BAM for folding into the OM. Importantly, in this structure SurA is stably bound to POTRA-1 of BAM in the absence of trapping via a disulphide bond at that location. Collectively these results show that an OMP can be simultaneously bound to BamA and SurA, and suggests that Extended SurA could remain bound to BAM at least during the early stages of OMP folding and insertion into the OM. In the second class of particles, no extra β-strands were visible at the BAM lateral gate, so we attribute this structure to the non-disulphide bonded complex observed biochemically after purification (Supplementary Fig. 8b). In this structure SurA is once again in the Extended conformation, as observed in the Arrival Complex, but the chaperone is not as well resolved (Supplementary Fig. 8f), presumably since it is not stabilised by a disulphide bond in this complex. This structure together with the Arrival Complex suggests that SurA binds POTRA-1 first prior to the OMP β-signal engagement with BamA β1.

### The release complexes
The SurA-OmpX construct disulphide bonded at BamA-β1 may have captured an early folding intermediate of OmpX on BAM, but OmpX cannot complete its journey into the membrane, because the construct is also concatenated to the C-terminus of SurA. To trap a late-stage folding intermediate of an OMP during delivery from SurA, an additional linker sequence would be needed to make the construct long enough to allow OmpX to complete its folding into the OM. Given the extensive literature on the creation of late stalled complexes on BAM in the absence of SurA, we switched the OMP client to EspP which has already been widely investigated structurally on BAM[19,21], with the added advantage that the necessary linker is an intrinsic part of EspP's sequence (Fig. 6a, Supplementary Fig. 9)[19,21]. EspP is a 12-stranded autotransporter and its sequence contains an additional 76 residues N-terminal to its β-barrel which form an α-helix that resides within the lumen of the EspP barrel in its native state[47]. These 76 residues provide the additional linker length we hypothesise is needed to enable folding to complete whilst the OMP remains bound to SurA at POTRA-1. Complexes of BAM disulphide bonded to SurA-EspP cross-linked via BamA β1 (S425C) and residue S1299C in the C-terminal β-signal of EspP were purified from the *E. coli* OM, and the structures solved by cryoEM. The structures revealed BamA- EspP- hybrid barrels captured as late

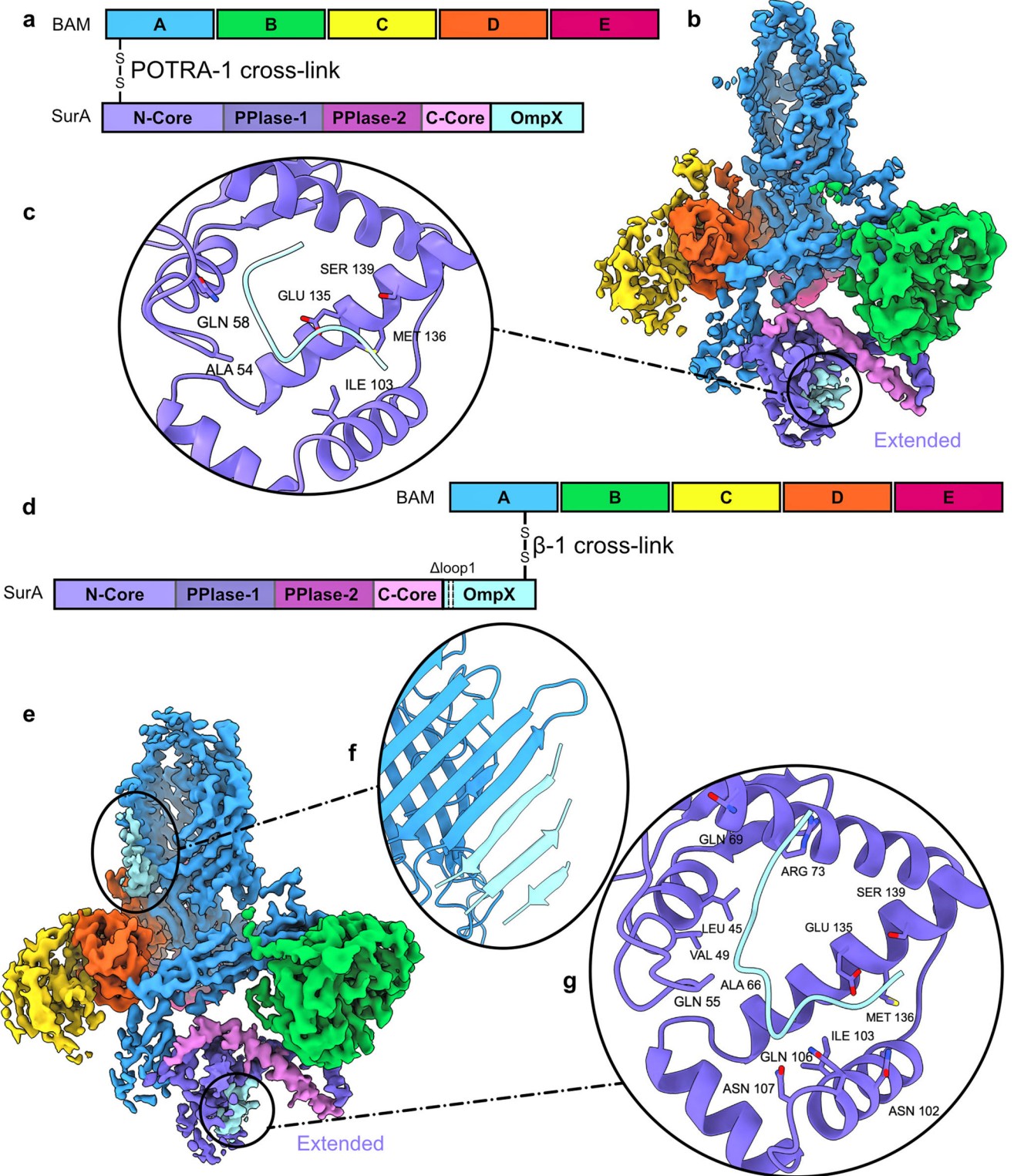

**Fig. 5 | The arrival and handover complexes of SurA-OmpX bound to BAM.**
**a** Schematic of SurA concatenated to OmpX and BAM trapped via a disulphide bond between POTRA-1 of BamA (R76C) and residue K27C of SurA (not to scale). Proteins and domains are coloured identically throughout. **b** CryoEM density map of the Arrival complex of Extended SurA-OmpX and Lateral Open BAM. Density corresponding to OmpX (cyan) is highlighted. **c** Zoom in of the density assigned to OmpX in the Core domain of SurA. **d** Schematic of SurA and BAM tethered via a disulphide bond between residue S425C in β1 of BamA and residue R170C which

lies in the C-terminal β-strand (β8) of OmpX. The C-terminus of SurA is also concatenated to the N-terminus of OmpX which has had loop 1 deleted (not to scale). **e** CryoEM density map of the Handover complex of Extended SurA-OmpX and Lateral Wide-open BAM showing three β-strands of the folding OmpX (cyan) that could be resolved in the structure. **f** Strands 8, 7 and 6 of OmpX folded at the lateral gate of BAM. **g** Zoom in of the density assigned to OmpX in the Core domain of SurA in this complex.

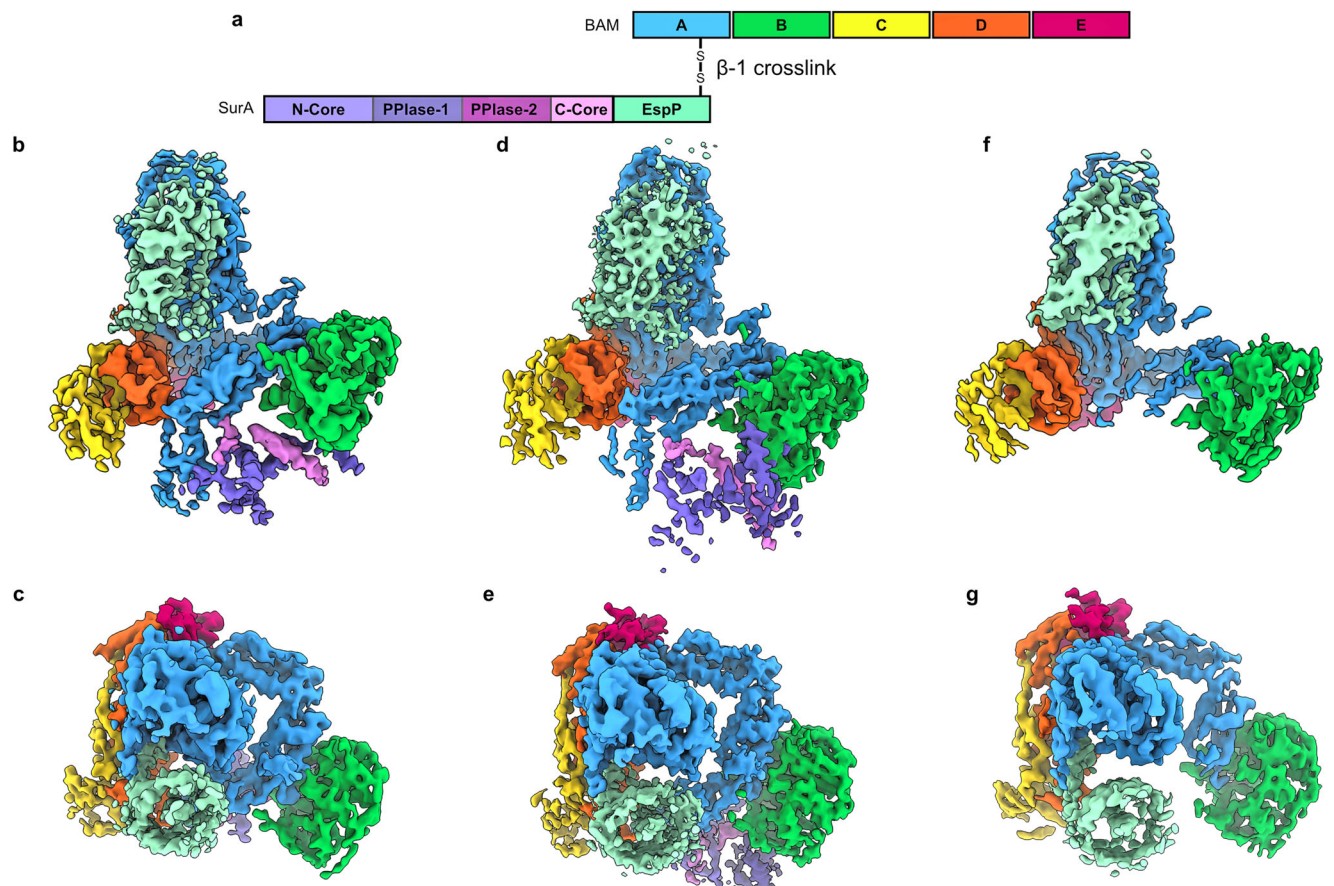

**Fig. 6 | A late stage folding intermediate of BAM: SurA-EspP. a** Schematic of proteins used to trap a late stage folding intermediate of EspP on BAM: SurA. The complex was purified directly from *E. coli* membranes. The location of cysteine residues on β1 of BamA (S425C) and β12 of EspP (S1299C), the latter of which is concatenated to the C-terminus of SurA via its natural 76 residue N-terminal linker, are shown (not to scale). **b** CryoEM density map of Extended SurA bound to Lateral

Open BAM with hybrid EspP barrel from side **c.** and top. **d** CryoEM density map of Compact SurA bound Lateral Open BAM with hybrid EspP barrel from side **e.** and top. **f** CryoEM density map of Lateral Open BAM with hybrid EspP barrel (no SurA density resulting in the loss of density for POTRA-1 and POTRA-2) from side **g.** and top.

stage folding intermediates, as expected[19,21]. Fully-folded EspP β-barrels that contain the native α-helix in the barrel's lumen were present in all structures (Fig. 6b–g). In these structures, both Extended and Compact conformations of SurA are present (Fig. 6b, d), as well as a structure in which SurA was not observed (Fig. 6f), consistent with the chaperone dissociating from POTRA-1 in these late-stage folding complexes. In the Extended SurA structure, no additional density for unfolded EspP was observed on SurA. Collectively, these results suggest that these late-stage folding intermediates of EspP are no longer being chaperoned by SurA. Indeed, dissociation of the substrate OMP from the chaperone would be a pre-requisite for release of the OMP from BAM and insertion into the OM. Hence, we term these as 'Release Complexes', in which SurA has returned to a structural ensemble that includes both Extended and Compact states. These Release Complexes also support the functional relevance of our previous POTRA-1:SurA disulphide bonded structures, since the interaction between POTRA-1 and SurA is again observed without SurA being covalently tethered to BAM at this position. This further supports that the POTRA-1:SurA Core domain interaction is a major binding mode of SurA on BAM.

## Discussion

How OMPs are efficiently folded and inserted into the OM poses a major challenge that must be overcome for successful bacterial growth and virulence. Indeed, dysregulation of OMP biogenesis results in the induction of OM stress responses and lability of the usually

impenetrable OM[1]. Accordingly, inhibiting BAM via antibody or anti-biotic binding is bactericidal[15,39,40,43,48]. However, it remains unknown how chaperone binding and OMP delivery to BAM are coordinated and controlled in the absence of an obvious energy source (the periplasm is devoid of ATP). Recent developments in cryoEM, combined with elegant experiments using disulphide bond trapping, have revealed structures of BAM in the act of folding OMPs at various stages of their assembly (Supplementary Fig. 1)[13,19–21]. However, these structures lack details of the first key step in OMP folding, namely their delivery by SurA to BAM.

Across six different cryoEM datasets, we describe structural details for putative OMP delivery and folding by BAM-SurA encompassing the Wait, Arrival, Handover and Release complexes (Fig. 7). We show, via disulphide bond mapping, proline substitutions and cryoEM maps of complexes purified directly from the OM, that SurA appears to bind to the BAM complex via a β-augmentation interaction. This interaction is mediated by an edge strand of POTRA-1 of BamA and N-terminal residues 23–28 of SurA, consistent with previous work and AlphaFold2 predictions[24,33]. We further show that the same binding interface between the proteins is formed both in the presence of the POTRA-1 disulphide (Wait and Arrival complexes (Fig. 7a, b)) and in its absence (Handover and Release complexes (Fig. 7c, d)). Disrupting this interaction interface via single proline mutants, or deletion of the identified 6-residue sequence of SurA, results in defects in OMP bio-genesis and cell envelope integrity. The phenotype for these variants mirrors deletion of the entire *surA* gene, which includes depletion of

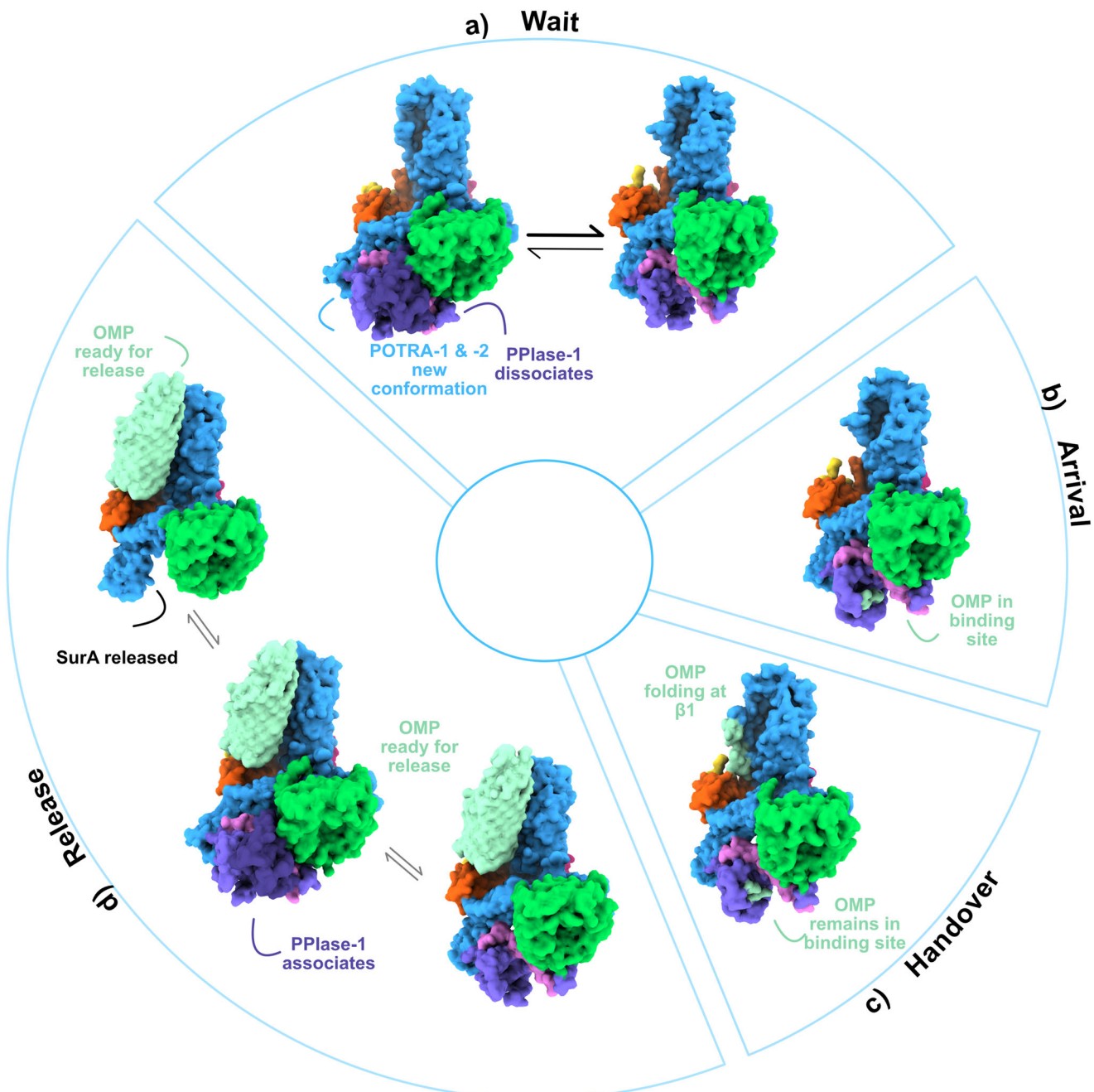

**Fig. 7 | Proposed model of OMP assembly orchestrated by BAM-SurA complexes. a** In the absence of an OMP, SurA can bind to BAM initiating pre-organisation of both binding partners. PPIase-1 of SurA dissociates from Core and POTRA-1 and -2 of BamA move closer to the BamA β-barrel. **b** Upon binding of an OMP client, SurA is found only in its Extended state with the OMP bound (at least in part) in its Core domain binding site. **c** During handover to β1 of BamA the OMP is simultaneously bound to both SurA and BAM. **d** Once folding is complete and the OMP released from SurA, PPIase-1 re-associates with Core and SurA can be released from BAM. BAM and SurA are then ready for a new folding cycle.

the major structural OMPs, OmpC and OmpF, as well as LptD which is involved in lipopolysaccharide production and is essential for bacterial growth[2]. Most notably, we show that removing the ability of SurA to bind BAM, by deletion of residues 23–28, results in OM defects, but induces a less broad stress response than deletion of the entire protein[49]. This is, in our opinion, powerful evidence that the chaperone and OMP delivery activities of SurA reside in different portions of the SurA molecule. Deleting just six N-terminal residues of SurA that are disordered in the crystal structure does not remove its general chaperone function, but specifically targets its ability to deliver OMPs to BAM for folding into the OM. Given the essentiality of BAM's activity

for OMP folding into the OM[6,7,35], these results imply that other routes for OMP delivery to BAM may be operative in the periplasm[3].

In the absence of a bound OMP, binding of SurA to BAM results in a concerted modulation of the conformation of both binding partners, as supported by the Wait Complexes described here. This change in conformation encompasses the dissociation of PPIase-1 from the Core domain of SurA, and a reorganisation of the interactions between POTRA-2 and BamD. This two-way communication exposes the client binding site of SurA[30,44] which lies between PPIase-1 and Core, and could serve to prime BAM to receive a substrate. The Lateral Closed conformation of BAM is likely to be required for initial recognition of

the OMP β-signal at BamA-β1. Structures with DAR-B, which functionally mimics an incoming β-signal, show a complex between SurA and Lateral closed BAM is formally possible (Fig. 2h, i). In turn, this suggests that binding of SurA to BamA POTRA-1 can precede substrate recognition by BamA-β1. The existence of the Wait Complex suggests that SurA may play different roles in the cell. SurA molecules may recognise and bind OMPs as they emerge into the periplasm from the Sec translocon, while other chaperone molecules might remain bound to BAM without a pre-bound client OMP, ready to accept one from a preloaded SurA-OMP complex. Successive handover of OMPs between chaperones during assembly has been suggested previously[25] and used to rationalise efficient chaperoning by SurA despite its weak affinity for its clients ($K_d$ ~µM). Such a scenario is also consistent with previous observations that multiple SurA molecules can bind a single OMP chain[44].

The binding of an unfolded OMP to SurA does not seem to generate novel conformations of BAM, as exemplified by the Arrival complex wherein the Extended SurA conformation predominates and BAM is in a Lateral Open state. It appears to be the OMP, rather than SurA, that drives the BamA lateral gate towards a Lateral Wide-open conformation, as observed in the Handover complex. Our data indicate that SurA can remain bound in the early stages of OMP folding. However, it is unclear at which stage of the BAM folding cycle in vivo SurA might bind POTRA-1, or dissociate from BAM and release its bound OMP clients. Further work is required to elucidate the precise order of events and the coordination between BAM and SurA during handover of a substrate OMP as folding and insertion into the OM progresses. We posit that SurA may remain bound to BAM for a complete folding cycle so as to enable vectorial transfer of the OMP substrate into the membrane. Such a scenario would enable SurA to prevent collapse of the unfolded OMP substrate that would slow or prevent successful folding, poisoning BAM by blocking its interaction surfaces for future folding events. Indeed, slowing the folding of OMPs on BAM is bactericidal[50].

Our structures support that the Core domain of SurA is essential both for its initial interaction with BAM (in the Wait Complexes; Fig. 7a), and its continued engagement with BAM throughout OMP folding, as seen in the Arrival (Fig. 7b) and Handover Complexes (Fig. 7c). This agrees with studies which have shown that the Core domain alone and full length SurA have a similar affinity for BAM[24], and that the Core domain alone is capable of complementation of membrane sensitivity of ΔsurA strains[51]. However, the roles of the PPIase domains in OMP folding remain more elusive. Deletion of PPIase-1 has no effect on tOmpA (the β-barrel domain of OmpA) folding kinetics in vitro, but dissociation of PPIase-1 from the Core domain is crucial for OMP assembly in vivo[24,31]. The intrinsically dynamic nature of SurA, as revealed here and elsewhere by smFRET, MD and other studies[30,46,52], allows the chaperone to sample the Extended conformation, which is then conformationally selected when bound by an OMP client or to BAM. A variant of SurA bearing the mutation S220A was previously discovered as a suppressor of BamB knockouts, and shown to exist only in the Extended conformation. This suggests that BamB may be involved in this conformational selection of Extended SurA in agreement with our structures[26,31,46,53]. Deletion of PPIase-2 has no effect on the affinity of SurA for BAM[24], consistent with our structures wherein no high-resolution density for PPIase-2 is present in any condition. A previous report showed that deletion of PPIase-2 reduces the BAM-catalysed folding rate of tOmpA, whilst AlphaFold2 predictions and co-evolution studies suggest that PPIase-2 interacts with BamE[24]. Hence, the PPIase-2 domain may optimise, or fine-tune, folding efficiency on BAM by transiently interacting with BamE[23]. Further work will be needed to test these hypotheses.

Together with previous studies of BAM-OMP interactions[13,19–21,47], the structures presented here describe the potential stages of an OMP's journey from its initial capture by SurA, through its vectorial delivery to BAM for templated folding via BamA-β1, to the release of the fully folded OMP β-barrel into the OM. Throughout its journey, SurA could remain bound to BAM, chaperoning its client and coordinating its delivery to BAM until folding is complete. This work has focused on 1:1 complexes between BAM and SurA for structural biology, but the periplasm is a complex, crowded environment. SurA is thought to be the major periplasmic chaperone for OMP biogenesis but it is not essential for growth in *E. coli*, so further work is required to understand the interplay and dynamics between BAM, OMPs and a plethora of periplasmic chaperones and/or proteases such as SurA, Skp, DegP or BepA, that are needed to fold an OMP into the OM[22]. Nonetheless, our results suggest that the interaction between SurA and BamA POTRA-1 is a crucial protein: protein interaction in OMP folding, the disruption of which erodes the barrier function of the OM. Given the proven efficacy of β-strand mimics in inhibiting BAM functions at other sites, such as darobactins and dynobactins, the interaction between BAM and SurA represents another target for the development of new antibiotics that focus specifically on chaperone-mediated delivery of OMPs to BAM.

## Methods

### Strains and plasmid construction

Details of plasmids, primers and strains used in this study are given in Supplementary Tables 3, 4 and 5. For pSCRhaB2-SurA(PelB-NTTS-TEV) a fragment containing a PelB signal sequence, N-terminal TwinStrep tag, TEV cleavage site, and the mature sequence of SurA was synthesized by GeneWiz (Germany) and subsequently moved into a pSCRhaB2 vector using restriction-ligation cloning. For pET28a-PelB-NTTS-TEV-SurA_OmpX, OmpX was amplified out of *E. coli* BW25113 with flanking cut sites and cloned into pSCRhaB2-SurA(K27C) linearized with the same cut sites at the C-terminus of SurA leaving small scar site between SurA and OmpX. This gene was then moved into pET28a using restriction-ligation cloning. To test multiple stalling and cysteine variants a pET28a-PelB-NTTS-TEV-SurA GoldenGate drop-in vector series was created to insert substrate genes at the C-terminus of SurA(K27C) separated by a GGGS linked. Substrate genes were synthesized by Twist Biosciences (San Francisco, USA) as gene fragments with BsaI sites for GoldenGate cloning. All site-directed mutagenesis was performed using Q5 site-directed mutagenesis (NEB).

### Design of disulphide trapped variants of BAM-SurA complexes

The sequence of mature SurA (residues 21-427) and BamA POTRA domains (residues 21-424) from *E. coli* K-12 were submitted to the Google Colab servers for AlphaFold2 from Google DeepMind and ColabFold accessed on 29/07/2021 (now at github.com/sokrypton/ColabFold). Predictions were manually inspected in PyMol2.0 and used to rationally design pairs of disulphides that would lock the complex in place. The PDB format file for the best scoring prediction is available at the DOI described in the Data Availability statement below. AlphaFold2 prediction of the full complex of SurA-BamABCDE (SurA-BAM) was described previously[24,33] and is also deposited at the same DOI.

### Co-expression tests of disulphide locked BAM-SurA

*E. coli* BL21(DE3) were co-transformed with plasmids pTrc99a-BamABCDE_CT8His (expressing the BAM complex with a His$_8$ tag on BamE[54]) and pSCRhaB2-SurA_NT-TwinStrep (expressing SurA with an N-terminal Twin-Strep tag) containing relevant cysteine variants. Overnight starter cultures of co-expression strains were inoculated into 10 ml of LB in 25 ml glass culture tubes pre-heated to 30 °C. Cultures were then grown and protein expression induced as described below except that induction was continued for 1 h. Cells were pelleted at 3200 x *g*, 5min, 4 °C, resuspended in 10ml chilled PBS + 0.2 mM 4,4′dipyridyldisulfide (4-DPS) and mixed gently at 4 °C for 30 min. The optical density at 600 nm ($OD_{600}$) was then measured and an equivalent number of cells as 1ml of $OD_{600} = 1.0$ was pelleted at 3200 x *g*,

15 min, 4 °C in a 1.5 ml microfuge tube. The supernatant was discarded and the pellet stored at −20 °C or used immediately. Pellets were resuspended in 100 μL of 1X SDS loading buffer (50 mM Tris-HCl pH6.8, 1.5% (w/v) SDS, 10% (v/v) glycerol, 0.1% (w/v) bromophenol blue), boiled for 10min, then analysed by western blotting.

## Western blots

4−20% Mini-Protean TGX SDS-PAGE gels (Bio-Rad) gels were run at 200 V. Gels were transferred to 0.2 μm PVDF membranes (Bio-Rad) by semi-dry transfer using the 'TGX' setting on a Trans-Blot Turbo system (Bio-Rad). Membranes were blocked for 30 min at room temperature (RT) with 5ml PBST (1X PBS, 0.1% v/v Tween-20) supplemented with 2% (w/v) skimmed milk powder, 1° antibody was added to this and incubated for a further 90 min at RT (αStrep 1:1000 – Qiagen, αHis 1:1000 – Merck, αSurA 1:1000, or αBamA 1:2500[47,55]). Membranes were then rinsed 3X in PBST, incubated for 1 h at RT in 5 ml PBST + 2% (w/v) milk powder + HRP-conjugated 2° antibody (GoatαRabbit – abcam, or Hamster αmouse – Cell signaling Technology), and rinsed 1X in PBST. SuperSignal western pico chemiluminescent substrate (Thermo-Fischer Scientific) was added to membranes and blots imaged on a Uvitec Alliance Q9 imaging system.

## In vivo complementation assay

BamA depletion strain JCM166[35] and SurA deletion strain AR208[36] were transformed with plasmids pZS21-BamA[37] and pZA31-SurA, respectively. For BamA complementation studies, colonies were inoculated into 10 ml LB supplemented with 0.1% (w/v) arabinose (Sigma Aldrich) + 50 μg/ml kanamycin (omitted for untransformed controls) and grown overnight at 37 °C in 25 ml glass culture tubes shaking at 220 rpm. The following day the $OD_{600}$ of each culture was measured and 10 ml pre-heated fresh LB + supplements in 25 ml glass culture tubes was inoculated to a starting $OD_{600}$ of 0.05. Cultures were then grown at 37 °C, 220 rpm to an $OD_{600}$ of ~ 0.6. Culture ODs were then normalized to $OD_{600}$ = 0.1 and serial dilutions made into LB only. 2 μL of each of these dilutions was then spotted onto LB + 1.5% (w/v) agar plates containing 0.1% (w/v) glucose (Fisher) and either 100 μg/ml or 0 μg/ml vancomycin (formedium). These were allowed to dry and the plate then incubated overnight at 37 °C. Plates were scored for growth and imaged in a Uvitec Alliance Q9 imaging system. For SurA complementation studies, the protocol was the same except that bacterial growth was supplemented with 50 μg/ml kanamycin and 25 μg/ml chloramphenicol (omitted for untransformed controls). These bacteria were plated onto LB + 1.5% (w/v) agar containing either 50 μg/ml or 0 μg/ml vancomycin.

## Sample preparation for proteomics

Single colonies from the Δ*surA* strain AR208, Δ*surA* strain complemented with pZA31-SurA (JEH108), or pZA31-SurAΔ23-28 (JEH199) were each inoculated into 10 ml LB in 25 ml glass culture tubes supplemented with 50 μg/ml kanamycin and 25 μg/ml chloramphenicol (JEH108 and JEH199) or kanamycin only (AR208) and grown overnight at 37 °C, 220 rpm. For each biological repeat, different colonies were picked. The following day the $OD_{600}$ was measured and samples were inoculated into 10 ml pre-heated fresh LB + 50 μg/ml kanamycin to a starting $OD_{600}$ of 0.05. Cultures were grown to an $OD_{600}$ of ~ 0.8 and an equivalent number of cells as 1 ml of $OD_{600}$ = 1.0 was pelleted at 3000 x *g*, 3 min, 4 °C in a 1.5 ml microfuge tube, the supernatant discarded, and the pellet stored at − 20 °C. Pellets were resuspended in 100 μL 1X lysis buffer (5% (w/v) SDS, 50 mM TrisHCl pH8.5) and then sonicated in a bath for 5 min. The sample was then centrifuged at 16,000 x *g*, 5 min, RT to pellet any cell debris and the supernatant retained. 20 μL of each sample was used to measure the total protein concentration by Bradford assay using the Pierce Detergent Compatible Bradford Assay Reagent. All samples were then normalized to 900 μg/ml in 100 μL (90 μg total protein in each sample) for MS analysis.

## Proteomics

Sample preparation of lysates for mass spectrometry (MS) analysis was performed using S-Trap micro spin columns (Protifi) according to the manufacturer's instructions. Briefly, reduction was performed by adding 20 mM dithiothreitol (10 min, 50 °C), followed by alkylation with 40 mM iodoacetamide (30 min, 20 °C). Samples were acidified by adding phosphoric acid (5% final concentration), and then diluted with 90% Methanol in 100 mM triethylammonium bicarbonate (TEAB) pH 7.1 (1:7 (v/v) sample: buffer). Trypsin (1 μg, Promega, UK) was added and the proteins were trapped on the S-trap column. The column was then washed three times with 90% Methanol in 100 mM TEAB pH 7.1. Trypsin solution (30 μL, 0.02 μg/uL trypsin) was added to the column which was then incubated for 90 min at 47 °C. Peptides were recovered by washing the column sequentially with 50 mM TEAB (40 μL), 0.2% (v/v) formic acid (40 μL), and 50% acetonitrile/ 0.2 % (v/v) formic acid (40 μL). The eluate was then evaporated to dryness in a vacuum centrifuge and the peptides resuspended in 0.1% (v/v) formic acid (20 μL) prior to MS. Peptides (5 μL) were injected onto a Vanquish Neo LC (Thermo Fisher Scientific) and the peptides were trapped on a PepMap Neo C18 trap cartridge (Thermo Fisher Scientific, 5 μm particle size, 300 μm x 0.5 cm) before separation using an Easy-spray reverse phase column (Thermo Fisher Scientific, 2 μm particle size, 75 μm × 500 mm). Peptides were separated by gradient elution of 2–40% (v/v) solvent B (0.1% (v/v) formic acid in acetonitrile) in solvent A (0.1% (v/v) formic acid in water) over 2 h at 250 nL.min⁻¹. The eluate was infused into an Orbitrap Eclipse mass spectrometer (Thermo Fisher Scientific) operating in positive ion mode. Data acquisition was performed in data dependent analysis (DDA) mode and fragmentation was performed using higher-energy collisional dissociation (HCD). Each high-resolution full scan (m/z 380–1400, R = 60,000) was followed by high-resolution product ion scans (R = 30,000), with a normalised collision energy of 30%. A cycle time of 3 s was used. Data were analysed using MaxQuant (v2.4.2.0) and Perseus (v2.0.10.0). Three technical replicates were measured for each condition. Search parameters include: sequence database = E. coli K12 proteome; digestion specificity = trypsin (K/R) with max 2 missed cleavages; fixed modifications = cysteine carbamidomethylation, variable modifications = methionine oxidation, minimum peptide length = 7 residues, precursor mass error tolerance = 8 ppm; MS/MS mass error tolerance = 20 ppm; peptide- and protein-level FDR = 0.01; minimum number of unique peptides for protein identification = 1. Statistical significance was determined using the ANOVA test implemented in Perseus and a false discovery rate of 0.05.

## Expression and purification of BAM-SurA complexes and BAM-SurA-substrate fusions

Two strategies were used for purification of complexes for cryo-EM and cross-linking studies. For BAM-SurA$_{POTRA1}$ cross-linked samples, *E. coli* BL21(DE3) was co-transformed with plasmids pTrc99a-BamA(R76C) BCDE$_{CT8His}$ and pSCRhaB2-SurA(K27C)$_{NT-TwinStrep}$. Overnight cultures were grown at 30 °C and the next day were diluted to a starting $OD_{600}$ of 0.05 in 10x1L of LB (Fisher) in 2L baffled flasks supplemented with 100 μg/ml carbenicillin (Formedium) and 10 μg/ml trimethoprim (Sigma Aldrich). Cultures were grown at 30 °C, 220 rpm, until reaching an $OD_{600}$ of ~ 0.6–0.8. Protein expression was then induced with 0.4 mM IPTG (Formedium) and 0.4% (w/v) L-rhamnose (Apollo Scientific), allowed to grow for a further 1.5 h, and then harvested by centrifugation at 7000 x *g*, 4 °C, 5 min in a JLA8.1000 rotor. Cells were pooled and resuspended in 1L of pre-chilled PBS (OXOID) before 0.2 mM 4-DPS (Sigma Aldrich) was added from a 20 mM stock in DMSO and incubated with stirring at 4 °C for 1 h. Cells were pelleted again by centrifugation at 7000 x *g*, 4 °C, 10 min and stored −20 °C.

For substrate-containing complexes *E. coli* Lemo21(DE3) cells were co-transformed with pTrc99a-BamA(S425C)BCDE$_{CT8His}$ and

pET28a-SurA_[substrate] containing relevant cysteine mutants. The protocol to purify the complex was carried out as the above except for the following: LB was supplemented with 50 μg/ml kanamycin (Formedium), 100 μg/ml carbenicillin, 25 μg/ml chloramphenicol (Formedium) and 0.5 mM L-rhamnose; protein expression of both plasmids was induced with 0.4 mM IPTG.

For purification of the complexes all buffers were pre-chilled and samples kept on ice or at 4 °C unless otherwise stated. Frozen cell pellets were resuspended in 100ml 20 mM Tris-HCl pH7.5 (Fisher), 1mM EDTA (Acros Organics) + EDTA-free protease inhibitor tablets (Merck), homogenized at 8000 rpm (), and then disrupted by 2 passes at 30 kPSI in a cell disruptor (Constant Systems). The cell lysate was centrifuged at 20,000 x g, 15 min, 4 °C in a JA25.50 rotor and the supernatant then subjected to centrifugation at 210,000 x g, 30 min, 4 °C in an MLA-50 rotor. The supernatant was discarded, and membrane pellets rinsed with 20 mM TrisHCl pH7.5, 1mM EDTA, before resuspension in 60 ml 20 mM Tris-HCl pH7.5, 150 mM NaCl, 1 mM EDTA, 1% (w/v) DDM (Melford) and disruption of membranes by 30 strokes in a Dounce homogeniser. The sample was then incubated on a roller at 4 °C for 1 h, followed by centrifugation at 210,000 x g, 4 °C, 30 min. The supernatant was filtered using 0.22 μm membrane (Merck), then loaded onto a 1ml StrepTrap HP Column (Cytiva) equilibrated with Strep Wash Buffer (20 mM Tris-HCl pH7.5, 150 mM NaCl, 0.05% (w/v) DDM) at ~1.5 ml.min⁻¹ via peristaltic pump. The column was washed with 10 colume volumes (CV) of Strep Wash Buffer, then eluted into 1 ml fractions using Strep Wash Buffer supplemented with 5 mM desthiobiotin (Merck). Protein-containing fractions as determined by $A_{280}$ were pooled and further purified using gel filtration on a Superdex 200 16/600 column on an ÄKTA protein purification system at 1 ml.min⁻¹ at RT in 20 mM Tris-HCl pH8.0, 150 mM NaCl, 0.02% (w/v) DDM. Fractions were assessed by western blotting for the presence of BAM and SurA/SurA-substrate and to assess the presence of disulphide cross-links. Complex-containing fractions were pooled and concentrated to ~2–4 mg/ml⁻¹ using Vivaspin 20 and Vivaspin 500 100 kDa MWCO centrifugal concentrators (Sartorius) and either put directly onto grids for cryo-EM analysis or snap frozen in liquid nitrogen and stored at −80 °C for later biochemical analyses. For Darobactin containing samples, darobactin-B (purified as described[15]) was added to the BAM-SurA sample at a 2x molar excess of darobactin-B. For WEYIPNV containing samples, WEYIPNV (Genscript) was added to the BAM-SurA sample at 10x molar excess of WEYIPNV.

### CryoEM grid preparation
Samples were applied to R1.2/1.3 (300 mesh) Quantifoil grids, previously plasma cleaned for 60 s using a Tergio Plasma Cleaner (PIE Scientific). Grids were blotted for 5-6 sec at 4 °C and >90% humidity and plunge-frozen in liquid ethane with a Vitrobot Mark IV 480 (ThermoFisher).

### CryoEM data collection and processing
Datasets were collected on a 300 keV Titan Krios electron microscope (ThermoFisher) in the Astbury Biostructure Laboratory operated with a Falcon4/Falcon4i detector in counting mode. POTRA-1 cross-link, POTRA-1 cross-link + Darobactin-B, β1 cross-link OmpX hybrid, β1 cross-link EspP hybrid datasets were collected using a Selectris energy filter operating with a 10 e-V slit. All datasets were collected with nominal defocus range of −0.9 to −3.0 μm. Data acquisition parameters for each dataset can be found in Supplementary Table 1. All processing was performed using Relion4[56], unless otherwise stated. EER format micrographs were fractionated with 0.8–0.9 e-/Å²/frame, motion corrected and dose-weighted. CTF parameters were estimated with CtfFind4.1[57]. Particles were initially picked with crYOLO (v1.8) using the general model[58]. After multiple rounds of 2D classification, an initial model was generated and particles were subjected to 3D classification.

**POTRA-1 cross-link dataset.** A 3-class 3D classification of 167,132 particles was completed and classified based on the conformations of SurA. The two major classes corresponding to Compact SurA and Extended SurA with 133,734 and 60,508 particles respectively were refined and subjected to iterative rounds of CTF-refinement and Bayesian polishing to final resolutions of 4.1 Å and 4.2 Å, respectively.

**POTRA-1 cross-link + darobactin-B dataset.** 555,616 particles were subjected to a 4-class 3D classification. The two most abundant classes corresponded to the Barrel Closed with Extended SurA and Barrel Closed with Compact SurA with 164,058 and 153,636 particles respectively. These two classes were refined and subjected to iterative rounds of CTF-refinement and Bayesian polishing to final resolutions of 2.9 Å and 3.1 Å respectively. The Compact surA map showed poorer resolution in SurA and this was classified further without alignment using a mask around SurA yielding 4 classes differing only in the position of SurA and POTRAs 1 and 2 relative to the rest of the BAM complex. The third most abundant class corresponded to Barrel open BAM and this was classified further to yield 3 classes. One class of 47,673 particles of barrel open BAM with Extended SurA and one of 37,457 particles of barrel open BAM with Compact SurA. These open barrel classes were subjected to one round of Refinement yielding global resolutions of 7.8 Å and 7.1 Å but were not refined any further and are consistent with the structures solved in the first POTRA-1 cross-link dataset.

**POTRA-1 cross-link + WEYIPNV dataset.** A 3-class 3D classification of 285,312 particles was completed and the most abundant class of 212,826 particles was refined and subjected to iterative rounds of CTF-refinement and Bayesian polishing to final resolution of 3.8 Å.

**POTRA-1 cross-link OmpX hybrid dataset.** 220,791 particles were subjected to a 6-class 3D classification. 25% particles were iteratively refined and polished to a global resolution of 5.8 Å. A 3D classification without alignment with a mask around SurA was completed to improve the resolution in the OmpX density and the most abundant class of 31,118 particles was refined again to a final global resolution of 5.3 Å.

**β1 cross-linked OmpX hybrid dataset.** 1,973,311 particles were subjected to a 6-class 3D classification. The two most abundant classes were refined and subjected to iterative rounds of CTF-refinement and Bayesian polishing. The two classes correspond to BAM Lateral Wide-open with OmpX at β1 (final global resolution 4.2 Å) and Extended SurA and BAM Lateral Open with Extended SurA (final global resolution 4.0 Å).

**β1 cross-link EspP hybrid dataset.** An initial 3-class 3D classification of 645,271 particles was completed. The two BAM containing classes were further classified into 2 classes each. Each of these four classes was then used to train a model in CrYOLO and the particles repicked. After 2D and 3D classification to remove lower resolution particles, each class was iteratively refined and polished before one round of Non-Uniform refinement in CryoSparc[59]. Each final map was sharpened using PostProcess in Relion. The final maps corresponded to BAM with a hybrid barrel of fully folded EspP with Compact SurA (final global resolution 4.3 Å), Extended SurA (final global resolution 5.2 Å) and no SurA (final global resolution 4.4 Å).

### Model building
For both conformations in the POTRA1 cross-link dataset, an existing model for BAM (5LJO[10]) and SurA (1M5Y[27]). were rigid body docked into the two maps in ChimeraX[60] and domains not present removed to generate an initial model. Molecular dynamics flexible fitting (MDFF)[61] was used to flexibly fit each model into the maps. These models were refined using iterative rounds of real space refinement in PHENIX1.19[62]

and manual refinement in COOT[63] until satisfactory geometry and fit between model and map assessed using MolProbity[64]. Initial models were rigid body fit into maps in ChimeraX. The POTRA1 cross-link models were then used for all subsequent models built where either BAM or SurA exists in these conformations. 7NRI was used as an initial BAM Lateral Closed model. OmpX in the Arrival complex was manually built as a polyalanine chain in COOT. OmpX in the Handover complex was manually built as a polyalanine chain except for β strand 8 which was manually built according to the sequence. EspP from 7TTC and 3SLT were used as initial models. Most of the residues of the barrels of EspP were modelled with sidechains deleted due to lack of resolution in these regions of the maps. Iterative rounds of real space refinement in PHENIX1.19 and manual building in COOT were completed and assessed using MolProbility. Details of the initial models used for each structure and the model building statistics for all structures herein are shown in Supplementary Table 2.

## Expression and purification of WT BAM

pTrc99a-BamABCDE$_{CT8His}$ was transformed into BL21(DE3) cells and selected with 100 μg/ml carbenicillin. Single colonies were used to inoculate 4 x 50 ml 2TY (1.6% (w/v) tryptone, 1% (w/v) yeast extract, 0.5% (w/v) NaCl) in 250 ml baffled conical flasks which were cultured at 37 °C, 220 rpm overnight. These cultures were pooled and then inoculated into 10x 1L 2TY in 2 L baffled flasks supplemented with 100 μg/ml carbenicillin to a starting OD$_{600}$ of 0.05. Cultures were grown to an OD$_{600}$ of 0.6–0.8 before protein expression was induced with 0.4 mM IPTG and cultures were allowed to grow for a further 60–90 min. Cells were harvested by centrifugation at 7000 x g, 10 min, 4 °C in a JLA8.1000 rotor and the pellet stored at −20 °C. For purification all buffers were pre-chilled and samples kept on ice or at 4 °C unless otherwise stated. Pellets were resuspended in 100 ml 20 mM Tris-HCl pH 7.5 with EDTA-free protease inhibitor tablets, homogenised at 8000 rpm, then lysed twice at 30 kPSI in a cell disruptor. The lysate was centrifuged at 20,000 x g, 4 °C, 15 min in a JA25.50 rotor and the supernatant then centrifuged at 210,000 x g, 4 °C, 30 min in an MLA-50 rotor. Membrane pellets were rinsed with 20 mM Tris-HCl pH7.5 then resuspended in 60 ml 20 mM Tris-HCl pH7.5, 150 mM NaCl, 1% (w/v) DDM and broken up with 30 strokes in a Dounce homogenizer before incubating for 1 h with agitation at 4 °C. The sample was centrifuged again at 210,000 x g, 4 °C, 30 min in an MLA-50 rotor, the supernatant was retained and filtered at 0.22 μm before being doped with imidazole (Acros Organics) to a final concentration of 20 mM. The sample was loaded onto a 5 ml HisTrap FF column (Cytiva) using a peristaltic pump equilibrated with Wash buffer (20 mM Tris-HCl pH 7.5, 150 mM NaCl, 0.05% (w/v) DDM) supplemented with 10 mM imidazole at 8 ml/min. The column was washed with 5CV of Wash buffer supplemented with 10 mM imidazole before bound proteins were eluted into 2.5 ml fractions with 4CV of Wash buffer supplemented with 500 mM imidazole. The A$_{280}$ of fractions was measured and major protein containing fractions pooled, concentrated to ~3 ml in a 100 kDa MWCO Vivaspin20 centrifugal concentrator (Sartorius) load. Finally, the protein was loaded onto a Superdex 200 16/600 pg gel filtration column on an ÄKTA protein purification system. Size exclusion was performed at 1 ml.min⁻¹ and 1 ml fractions analysed by SDS-PAGE. BAM-containing fractions were pooled and concentrated as above to 30–50 μM before being snap froze in liquid nitrogen and stored at −80 °C.

## Expression and purification of SurA

This purification was described previously[24]. Briefly, pET28b-SurA$_{21-428}$NT$_{6His}$ were transformed into BL21(DE3) and the bacteria cultured at 37 °C to an OD$_{600}$ of 0.6 before inducing protein expression with 0.4mM IPTG at the lower temperature of 20 °C for 18 h, before cells were harvested by centrifugation. Cells were lysed using a cell disruptor, the lysate clarified, then applied to a 5 ml HisTrap FF column by

peristaltic pump. Columns were washed with 25 mM Tris-HCl pH 7.2, 150 mM NaCl, 20 mM imidazole then SurA denatured on-column with 25 mM Tris-HCl pH 7.2, 6M guanidine-HCl. SurA was renatured with a 25 mM Tris-HCl pH7.2, 150 mM NaCl wash, then eluted with 25 mM Tris-HCl pH 7.2, 150 mM NaCl, 500 mM imidazole. The eluate was dialysed against 25 mM Tris-HCl pH 8.0, 150 mM NaCl overnight and then His-tagged TEV protease (expressed and purified described previously[65]) and 0.1% (v/v) β-mercaptoethanol (β-ME) was added. After overnight incubation at 4 °C the sample was again loaded onto a 5 ml HisTrap FF column to remove the TEV protease and cleaved His-tag. Unbound flow-through containing SurA$_{21-428}$ (or variants) were dialysed against 25 mM Tris-HCl pH 8.0, 150 mM NaCl and then concentrated using 5 kDa MWCO Vivaspin20 centrifugal concentrators (Sartorius) to ~200μM, snap-frozen in liquid nitrogen and stored at −20 °C. For Cys variants (created previously[30]), 1mM DTT was added to all buffers not otherwise containing β-ME.

## Labelling of SurA cysteine variants for smFRET

Labelling was performed as described previously[30]. SurA double-Cys variants were buffer exchanged into 25 mM Tris-HCl pH 7.2, 150 mM NaCl, 5 mM DTT using a 0.5 ml 7 kDa Zeba-Spin desalting column (Thermo Fisher Scientific) and incubated at RT for 30 min to allow reduction of any aberrant disulphides. The sample was then buffer exchanged into 25 mM Tris-HCl pH 7.2, 150 mM NaCl, 1mM EDTA, adjusted to a protein concentration of 50 μM, and a 10-fold molar excess of Alexa Fluor 488 C5 maleimide and Alexa Fluor 594 C5 maleimide (Thermo Fisher Scientific) was added and incubated for 2 h at RT with gentle agitation. The reaction was quenched with a 10-fold molar excess (over the dye concentration) of β-ME. Protein was separated from unbound dye on a Superdex 200 10/300 GL column equilibrated with 50mM Tris-HCl pH 8.0, 150 mM NaCl and sample eluted with a flow rate of 0.5 ml.min⁻¹. Labelled protein-containing aliquots were combined, approximate labelled stoichiometries checked by A$_{280}$, A$_{495}$, and A$_{590}$ on a Nanodrop Spectrophotometer (Thermo Fisher Scientific). Samples were then aliquoted, snap frozen in liquid nitrogen, and stored at −80 °C.

## Single-molecule FRET data acquisition

Single molecule experiments were performed on a custom-built confocal epi-illuminated microscope in an inverted-stage configuration. PIE (pulsed interleaved excitation) at a frequency of 40 MHz was achieved using a 480 nm (PiLO48XSM, Advanced Laser Diode Systems) and 561 nm (PDL 800-D, PicoQuant) with average laser powers of 60 and 30 uW at the sample respectively[66]. The two lasers were combined using a dichroic mirror (Di02-R561-25 x 36, Semrock) before being coupled into a single mode fibre (P3-460B-FC-1, Thorlabs Inc.) using an achromatic collimator (PAF2A-A15A, Thorlabs Inc.). The laser emission from the fibre was then collimated (60FC-4-A11-01, Schäfter + Kirchhoff GmbH) and reflected from a dichroic mirror (Di03-R488/561-t1-25 x 36) into a 100x 1.45 numerical aperture oil-immersion objective (CFI plan apochromat lambda, Nikon Instruments). Light emitted from the sample was recollected by the same objective before being focused through a 100 μm pinhole (P100D Thorlabs Inc.) and collimated (AC254-050-A, Thorlabs Inc.). The emission light is then split into two using a dichroic mirror (Di02-R561-25 x 36, Semrock) before band-pass filters remove further excitation light in each channel (ET525/50m & ET605/52m, Chroma Technology GmbH). The two emission pathways are then focused by a lens (AC254-050-A, Thorlabs Inc.) onto two single-photon avalanche diodes (PD-100-CTD & PD-050-CTD, PicoQuant). Signal from the detectors and a sync signal from the laser was sent to a time-correlated single-photon counting data acquisition card (TimeHarp 260 PICO, PicoQuant). SymPhoTime64 software (PicoQuant) was used to acquire data. Labelled SurA was diluted to ~50 pM in 20 mM Tris-HCl pH 7.5, 150 mM NaCl, 0.05% (w/v) DDM and 1 mM 6-hydroxy-2,5,7,8-tetramethylchroman-2-carboxylic acid (Trolox) to

reduce dye blinking and bleaching[67]. In conditions containing BAM and WEYIPNEV they were added to 7.5 µM.

## Single-molecule FRET data analysis

PTU files from SymPhoTime64 software were converted into HDF5 data files using phconvert (https://github.com/Photon-HDF5/phconvert). These files were then analysed using the FRETBursts python package[68]. Firstly, the background was independently calculated for every 120 s period of measurement. Following the background calculation, a dual channel burst search was performed, selecting bursts with a minimum threshold of 6x the background signal in the donor and acceptor channels, a minimum total burst size of 100 photons in the donor and acceptor channels from the donor excitation and at least 15 photons in the acceptor excitation acceptor emission channel. Bursts were then corrected using the following values: donor leakage = 0.10, direct excitation = 0.12, gamma factor = 0.90 and beta factor = 0.30. Bursts were then filtered by removing events with an S > 0.7 and <0.3 to remove bursts with suspected photobleaching or blinking of the donor or acceptor.

The uncorrected bursts were then sectioned into 1 ms time bins and exported for fitting using photon distribution analysis (PDA)[69,70] using the PAM (PIE Analysis with MATLAB) software (https://gitlab.com/PAM-PIE/PAM)[71]. Settings for the PDA fitting was as follows: $R_0$ (Förster Radius) = 58.89 Å, minimum number of photons per bin = 30, maximum number of photons per bin = 200, number of bins in PR histogram = 100. An uncorrected stoichiometry threshold of 0.95 was applied to remove time bins suspected of acceptor photobleaching which were not removed in previous steps. Correction factors for the PDA fitting were kept the same as for burst selection apart from direct excitation, which in PDA requires the probability of direct acceptor excitation by the donor excitation laser which can be calculated from the absorption spectra and extinction coefficient of the dyes (for Alexa Fluor 488 and Alexa Fluor 594 the value is 0.077). The data sets for each FRET pair (i.e., Core: PPIase1 and Core: PPIase2) were fit independently. Within each dataset each condition (e.g., SurA, SurA+BAM) was fit globally. In this global fitting the mean distances of the distributions were shared globally across all conditions with the amplitudes of each component allowed to vary independently. Core: PPIase1 and Core: PPIase2 required 4 and 3 states to explain the data respectively. The width (δ) of the fitted distributions were fixed to a fraction (κ) of the mean distance (R) (i.e., $\delta = \kappa * R$). The fraction was optimised globally for each FRET pair during fitting (Core: PPIase1 κ = 0.087, Core: PPIase1 κ = 0.106).

## Reporting summary

Further information on research design is available in the Nature Portfolio Reporting Summary linked to this article.

## Data availability

CryoEM reconstructions and corresponding coordinates have been deposited in the Electron Microscopy Data Bank (EMBD) and the Protein Data Bank (PDB), respectively: Wait Complex Extended SurA (https://www.ebi.ac.uk/emdb/EMD-18035/, https://www.wwpdb.org/pdb?id=pdb_00008pz2), Wait Complex Compact SurA (https://www.ebi.ac.uk/emdb/EMD-18034/, https://www.wwpdb.org/pdb?id=pdb_00008pz1), Wait Complex with Darobactin Extended SurA (https://www.ebi.ac.uk/emdb/EMD-18046/, https://www.wwpdb.org/pdb?id=pdb_00008pzv), Wait Complex with Darobactin Compact SurA (https://www.ebi.ac.uk/emdb/EMD-18045/, https://www.wwpdb.org/pdb?id=pdb_00008pzu), Arrival Complex (https://www.ebi.ac.uk/emdb/EMD-18564/, https://www.wwpdb.org/pdb?id=pdb_00008qpw), Handover Complex (https://www.ebi.ac.uk/emdb/EMD-18563/, https://www.wwpdb.org/pdb?id=pdb_00008qpv), Release Complex Extended SurA (https://www.ebi.ac.uk/emdb/EMD-18543/, https://www.wwpdb.org/pdb?id=pdb_00008qp5), Release Complex Compact SurA (https://www.ebi.ac.uk/emdb/EMD-18053/, https://www.wwpdb.org/pdb?id=pdb_00008q0g) and Release Complex No SurA (https://www.ebi.ac.uk/emdb/EMD-18562/, https://www.wwpdb.org/pdb?id=pdb_00008qpu). The raw cryoEM datasets used in this study have been deposited to the Electron Microscopy Public Image Archive (EMPIAR): POTRA-1 Cross-link (https://www.ebi.ac.uk/empiar/EMPIAR-12197/), POTRA-1 Cross-link with darobactin (https://www.ebi.ac.uk/empiar/EMPIAR-11933/), POTRA1 Cross-link OmpX hybrid (https://www.ebi.ac.uk/empiar/EMPIAR-11939/), β1 Cross-link OmpX Hybrid (https://www.ebi.ac.uk/empiar/EMPIAR-11940/) and β1 Cross-link EspP Hybrid (https://www.ebi.ac.uk/empiar/EMPIAR-11941/). Details of all deposition codes are also available in Supplementary Table 2. The raw proteomics data have been deposited to the ProteomeXchange Consortium via the PRIDE[72] partner repository with the dataset identifier PXD046606[73]. The raw data for smFRET and histograms, and the PDB file for the AlphaFold2 prediction for the BAM-SurA complex, are available at the University of Leeds Data Repository: https://doi.org/10.5518/1460. All uncropped blots and gels and source data are provided in Source Data. Source data are provided with this paper.

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

## Acknowledgements

We thank Bob Schiffrin, David Brockwell, and all members of the Radford and Ranson labs for helpful discussions, and Nasir Khan for excellent technical support. CryoEM data were collected at the Astbury Biostructure Laboratory, funded by the University of Leeds and Wellcome (108466/Z/15/Z; 221524/Z/20/Z) and we thank facility staff for their technical input. The Mass Spectrometry Facility instrumentation used in this work was funded by Wellcome (223810/Z/21/Z). K.L.F. and J.E.H. acknowledge funding from the MRC (MR/P018491/1). K.L.F. is funded by the BBSRC (BB/X015653/1) and JAC by BBSRC (BB/T008059/1). SER holds a Royal Society Professorial Fellowship (RSRP/R1/211057). A.N.C. acknowledges the support of a Sir Henry Dale Fellowship jointly funded by Wellcome and the Royal Society (220628/Z/20/Z) and a University Academic Fellowship from the University of Leeds. T.F.S. and N.B. acknowledge funding from the German Federal Ministry of Education and Research (BMBF, via grant GBi2S and German Centre for Infection Research (DZIF) 09.918). The anti-BamA polyclonal antibody used for western blotting and pTrc99a-BamABCDE-CT8His plasmid was a kind gift from Harris Bernstein (NIH, USA). The anti-SurA polyclonal used for western blotting was a generous gift from Jean-François Collet (UCLouvain, Belgium). The BamA depletion strain (JCM166), SurA deletion strain (AR208), and BamA complementation vector (pZS21-BamA-NT6His) were generously provided by Tom Silhavy (Princeton, USA). All materials are available from the authors upon request.

## Author contributions

K.L.F. and J.E.H. contributed equally to this study and are designated as joint first authors. K.L.F. collected and processed all cryoEM data. J.E.H. led on construct design, molecular biology and protein production, assisted by R.J.H. Microbiology experiments were performed by KLF and JEH, as were proteomics experiments and data analysis (assisted by ANC). J.A.C. performed single-molecule fluorescence experiments and data analysis. N.B. and T.F.S. produced and purified darobactin-B. The paper was written and edited by all authors, and all authors contributed to experiment design. The project was supervised by S.E.R. and N.A.R.

## Competing interests

The authors declare no competing interests.
