## [Peer Review File · Nature Communications]

Outer membrane protein assembly mediated by BAM-SurA complexesEditorial Note: This manuscript has been previously reviewed at another journal that is not operating a transparent peer review scheme. This document only contains reviewer comments and rebuttal letters for versions considered at *Nature Communications*.

Reviewer #1 (Remarks to the Author):

I find the work presented in this manuscript, in which the authors dissect the process by which the periplasmic chaperone SurA delivers Outer Membrane Proteins (OMPs) to the β -barrel Assembly Machinery (BAM), to be of high quality. It will be of interest to the readers of Nature Communications. The authors have done a great job in addressing my comments.

Reviewer #2 (Remarks to the Author):

The authors have addressed my inquiries. However, I still believe it is inappropriate to conclude the interactive mechanism of proteins based on manually fused complexes. The observation that BamA crosslinks with either end of SurA-OmpX (POTRA1:surA or β 1:OmpX) shows different substrate binding states suggests that the fusion of SurA-OmpX may have introduced artificial constraints which could prevent the natural and simultaneous association of SurA and OmpX with BamA. Consequently, the uncrosslinked structure of SurA resulting from β 1:OmpX-SurA crosslinking might not accurately represent its physiological conformation.

It is also risky to propose the structure resulting from the fused and crosslinked complexes as stages of OMP assembly.

1. It is not very convincing that the SurA-crosslinked BAM structure represents a waiting status, as SurA is believed to sequester unfolded OMP in the periplasm to prevent aggregation. SurA is more likely to carry substrates to BAM rather than binding to BAM in an apo conformation to await substrate delivery. This SurA-bound conformation is likely an artifact of the disulfide bond rather than a physiological state.

The authors claim that the binding of the extended SurA brings POTRA1 and 2 closer to the BamA barrel. Could the authors discuss how this induced conformational change might affect the opening status of BamA barrel, as it appears that the conformations of the BamA barrel have remained the same in the two structures?

2. It is also hard to say whether the substrate binding details shown in the Arrival and Handover structures are physiological binding or due to protein fusion.

3. In the Release structures, it seems that the compact or extended SurA binds to the POTRA domains of BamA and BamB at different positions. It does not seem to represent consecutive conformational changes caused by association or dissociation of the P1 of SurA. Considering that SurA binds to BAM transiently with low affinity, it is more likely that by the time EspP is assembled into a barrel, SurA could have dissociated from BAM. Therefore, the simultaneous binding of the EspP barrel and SurA is likely due to the fusion, and the bound compact or extended SurA structures are likely randomly resulted as a fusion affixture.

I suggest the authors avoid making claims about the various stages of OMP assembly based on the disputable SurA-bound BAM structure unless the authors can prove that these structures truly represent physiological conformations without the influence of engineered tethering.

Reviewer #3 (Remarks to the Author):

The authors have done an excellent job addressing many of my concerns and improving their manuscript. However, as described below, some of my concerns have not yet been fully addressed. My primary concern is that some of the results are still overinterpreted. Regarding the authors' reply to my comment that their conclusions should be toned down because there are too many unnatural constraints in their system, I would note that, at least in my opinion, an experimental strategy cannot be justified simply because it has been used in several papers that were published in top journals. It could turn out that all of the papers are noteworthy, but all have similar flaws. Although I agree that the Nature paper by Tomasek et al is a seminal study, it is conceivable that deleting loop 1 in BamA sends the protein into a non-productive folding pathway and that the cryo-EM results do not show a true on-pathway folding intermediate.

Specific comments:

1) In my original review I focused on the fundamental problems of the authors' experimental approach and forgot to mention that Bennion D et al. 2010 Mol Microbiol 77:1153-71 published strong evidence that SurA binds to the first POTRA domain of BamA and that mutations in BamA that impair this interaction also impair OMP assembly. To be fair, I do not want to subject the authors to "double jeopardy" by bringing up a new issue, but for the sake of accuracy the authors should cite this paper in the Introduction (or possibly the first paragraph of the Results) and change line 36 in the Abstract to something like "Consistent with previous results, mutations that disrupt...". While they certainly have obtained significant new insights into the interaction between SurA and BamA they should acknowledge that they are not the first group to investigate this interaction.

2) Regarding my original comment 11: I still think that the authors should interpret the results they obtained with the SurA-OmpX fusion protein (Fig. 5) more cautiously. Because of the constraints imposed on the system (the crosslinking of BamA and SurA and the covalent link between SurA and OmpX), and the ability of OmpX to fold into a different conformation than it does under physiological conditions, it is not surprising that the OmpX β signal does not interact with BamA β 1. The authors should acknowledge the limitations of the data (perhaps around line 277) and their inability to determine if the structure shows events that occur under physiological conditions. The 'Arrival Complex' is a nice idea, but as I indicated previously it is impossible to determine from the data if SurA interacts with BamA POTRA1 before an OMP β signal interacts with BamA β 1 or if an OMP β signal interacts with BamA β 1 before SurA interacts with BamA POTRA1.

3) Regarding my original comment 12: Once again, the results should still be interpreted more cautiously by adding a caveat to the text (perhaps around line 300). As in the description of the 'Arrival Complex', the timing of events under physiological conditions cannot be discerned from the data. Does the OMP β signal bind to BamA β 1 and then begin to fold, or does it begin to form a β sheet while it is still bound to SurA? Furthermore, it is unclear if an OMP can be bound to both SurA and BamA simultaneously. The way the experiment was designed, OmpX binds to BamA because it was tethered by a disulphide bond, and OmpX might bind to the SurA core because its N-terminus was fused to the C-terminus of SurA and the protein does not have

many options.

4) Line 312: This new statement should be toned down. Instead of “captured an early folding intermediate” the authors should state something like “likely captured” or “appear to capture”.

5) Regarding my original comment 15: I still think that lines 338-340 should be modified. As I mentioned previously, the interaction between SurA and BamA POTRA1 is difficult to interpret because SurA has nowhere to go. It is not correct to state that “These Release Complexes reinforce the functional relevance of our previous POTRA-1:SurA disulphide bonded structures” because even without a disulphide bond SurA appears to be positioned to interact with POTRA1. The authors should either remove this sentence or modify it to indicate that the results are consistent with other data (e.g., the crosslinking data) that indicate that SurA binds to BamA POTRA1.

6) Fig. 7 and paragraph that starts at line 356: Because of the inherent limitations of the data, the authors should clearly state that the ‘Wait, Arrival, Handover and Release’ stages are a model, and not a sequence of events that has been clearly demonstrated by the results. The legend to Fig. 7 should also indicate that the authors are displaying a model.

7) Regarding my original comment 16: I do not think that the authors have addressed this concern adequately. The new statements on lines 375-377 and 438-443 are very vague and confusing, and do not incorporate their work into the context of the whole field. It would be fair to say that we know more about the environment of the periplasm than that it is complex and crowded. As I originally noted, there is a lot of evidence in the literature that specific chaperones such as Skp and BepA play important roles in OMP biogenesis, that many OMPs can be assembled at least to some degree without SurA (and possibly without any other chaperone), and that the interaction of SurA with BamA is “timed” so that defective OMPs can be removed by other factors. A new review by Devlin and Fleming (TIBS 2024, doi: 10.1016/j.tibs.2024.03.015) documents some of the unique and redundant properties of periplasmic chaperones that have been discovered. I am not suggesting that the authors write a paragraph about other chaperones, but I think they should at least add a couple of sentences that clearly state that SurA is not essential for OMP assembly, that SurA might not bind to BamA and a partially folded OMP indefinitely, and that specific chaperones have distinct functions in OMP biogenesis.

Minor comment:

1) Line 124: For clarity the authors should state that SurA residue 23 is the N-terminal residue of the mature protein because the first 22 residues are the cleaved signal peptide.

Response to Referees

NCOMMS-24-18096A; Fenn, Horne et al; "Outer membrane protein assembly mediated by BAM-SurA complexes"

REVIEWERS' COMMENTS

Reviewer #1 (Remarks to the Author):

I find the work presented in this manuscript, in which the authors dissect the process by which the periplasmic chaperone SurA delivers Outer Membrane Proteins (OMPs) to the β -barrel Assembly Machinery (BAM), to be of high quality. It will be of interest to the readers of Nature Communications. The authors have done a great job in addressing my comments.

We thank the referee for their positive response.

Reviewer #2 (Remarks to the Author):

The authors have addressed my inquiries. However, I still believe it is inappropriate to conclude the interactive mechanism of proteins based on manually fused complexes. The observation that BamA crosslinks with either end of SurA-OmpX (POTRA1:surA or β 1:OmpX) shows different substrate binding states suggests that the fusion of SurA-OmpX may have introduced artificial constraints which could prevent the natural and simultaneous association of SurA and OmpX with BamA. Consequently, the uncrosslinked structure of SurA resulting from β 1:OmpX-SurA crosslinking might not accurately represent its physiological conformation.

It is also risky to propose the structure resulting from the fused and crosslinked complexes as stages of OMP assembly.

1. It is not very convincing that the SurA-crosslinked BAM structure represents a waiting status, as SurA is believed to sequester unfolded OMP in the periplasm to prevent aggregation. SurA is more likely to carry substrates to BAM rather than binding to BAM in an apo conformation to await substrate delivery. This SurA-bound conformation is likely an artifact of the disulfide bond rather than a physiological state.

*There is no question here, but we note that this complex is made *in vivo* so while we don't know it is physiologically relevant it is physiologically possible. We do not believe that the referee has any evidence to assert the likelihood one way or the other. We also note that the manuscript now deliberately says 'suggests that SurA may play different roles in the cell' (line 395).*

The authors claim that the binding of the extended SurA brings POTRA1 and 2 closer to the BamA barrel. Could the authors discuss how this induced conformational change might affect the opening status of BamA barrel, as it appears that the conformations of the BamA barrel have remained the same in the two structures?

We are extremely reluctant to speculate on this matter because it is impossible to extrapolate allosteric mechanisms from static structures of protein complexes. It is not included in this manuscript because it is a negative result but we did look very carefully and there is no evidence of intermediate structures.

We also note that this is a very contentious issue in the field, and we think it unwise for both us and Nature Comms to speculate without any evidence one way or the other.

2. It is also hard to say whether the substrate binding details shown in the Arrival and Handover structures are physiological binding or due to protein fusion.

We agree that it is impossible to rule out that substrate binding details shown in the Arrival and Handover structures might not be physiological, but the SurA binding region we identify replicates the SurA binding site identified in literature. Equally the binding of OmpX to $\beta 1$ of BamA matches previous structures solved by cryoEM.

3. In the Release structures, it seems that the compact or extended SurA binds to the POTRA domains of BamA and BamB at different positions. It does not seem to represent consecutive conformational changes caused by association or dissociation of the P1 of SurA. Considering that SurA binds to BAM transiently with low affinity, it is more likely that by the time EspP is assembled into a barrel, SurA could have dissociated from BAM. Therefore, the simultaneous binding of the EspP barrel and SurA is likely due to the fusion, and the bound compact or extended SurA structures are likely randomly resulted as a fusion affixture.

The binding of compact and extended SurA to BamA and BamB observed in the $\beta 1$ -EspP dataset are identical to structures observed without a tethered substrate (POTRA1 cross-link and POTRA1 crosslink + darobactin B) and match the conformational changes caused by association/dissociation of PPlase-1. Our point is that SurA can have dissociated from BAM but also can be waiting at the bottom of BAM having returned to an equilibrium between compact and extended.

I suggest the authors avoid making claims about the various stages of OMP assembly based on the disputable SurA-bound BAM structure unless the authors can prove that these structures truly represent physiological conformations without the influence of engineered tethering.

We believe our manuscript is appropriately qualified. For example, line 308 'suggests that extended SurA could remain bound', line 390 'could serve to prime BAM', line 401 'consistent with previous observations' line 412 'SurA may remain bound to BAM', and line 443 'SurA could remain bound to BAM',

Reviewer #3 (Remarks to the Author):

The authors have done an excellent job addressing many of my concerns and improving their manuscript. However, as described below, some of my concerns have not yet been fully addressed. My primary concern is that some of the results are still overinterpreted. Regarding the authors' reply to my comment that their conclusions should be toned down because there are too many unnatural constraints in their system, I would note that, at least in my opinion, an experimental strategy cannot be justified simply because it has been used in several papers that were published in top journals. It could turn out that all of the papers are noteworthy, but all have similar flaws. Although I agree that the Nature paper by Tomasek et al is a seminal study, it is conceivable that deleting loop 1 in BamA sends the protein into a non-productive folding pathway and that the cryo-EM results do not show a true on-pathway folding intermediate.

Specific comments:

1) In my original review I focused on the fundamental problems of the authors' experimental

approach and forgot to mention that Bennion D et al. 2010 Mol Microbiol 77:1153-71 published strong evidence that SurA binds to the first POTRA domain of BamA and that mutations in BamA that impair this interaction also impair OMP assembly. To be fair, I do not want to subject the authors to “double jeopardy” by bringing up a new issue, but for the sake of accuracy the authors should cite this paper in the Introduction (or possibly the first paragraph of the Results) and change line 36 in the Abstract to something like “Consistent with previous results, mutations that disrupt...”. While they certainly have obtained significant new insights into the interaction between SurA and BamA they should acknowledge that they are not the first group to investigate this interaction.

We thank the referee for their comment and now reference Bennion et al. in line 85. We note however that the engineering completed in that paper is informative and relevant to cite but different to the studies described here.

2) Regarding my original comment 11: I still think that the authors should interpret the results they obtained with the SurA-OmpX fusion protein (Fig. 5) more cautiously. Because of the constraints imposed on the system (the crosslinking of BamA and SurA and the covalent link between SurA and OmpX), and the ability of OmpX to fold into a different conformation than it does under physiological conditions, it is not surprising that the OmpX b signal does not interact with BamA b1. The authors should acknowledge the limitations of the data (perhaps around line 277) and their inability to determine if the structure shows events that occur under physiological conditions. The ‘Arrival Complex’ is a nice idea, but as I indicated previously it is impossible to determine from the data if SurA interacts with BamA POTRA1 before an OMP b signal interacts with BamA b1 or if an OMP b signal interacts with BamA b1 before SurA interacts with BamA POTRA1.

We have added a sentence at line 314-315 to say that we are suggesting based on our data that SurA binds POTRA-1 first before OmpX binds to BamA β 1. We have also modified line 408-409 to now say that it is unclear at which stage of the BAM folding cycle SurA might bind POTRA-1.

3) Regarding my original comment 12: Once again, the results should still be interpreted more cautiously by adding a caveat to the text (perhaps around line 300). As in the description of the ‘Arrival Complex’, the timing of events under physiological conditions cannot be discerned from the data. Does the OMP b signal bind to BamA b1 and then begin to fold, or does it begin to form a b sheet while it is still bound to SurA? Furthermore, it is unclear if an OMP can be bound to both SurA and BamA simultaneously. The way the experiment was designed, OmpX binds to BamA because it was tethered by a disulphide bond, and OmpX might bind to the SurA core because its N-terminus was fused to the C-terminus of SurA and the protein does not have many options.

Please see final comment to reviewer 2.

4) Line 312: This new statement should be toned down. Instead of “captured an early folding intermediate” the authors should state something like “likely captured” or “appear to capture”.

We have amended the line accordingly (now line 318).

5) Regarding my original comment 15: I still think that lines 338-340 should be modified. As I mentioned previously, the interaction between SurA and BamA POTRA1 is difficult to interpret because SurA has nowhere to go. It is not correct to state that “These Release Complexes reinforce the functional relevance of our previous POTRA-1:SurA disulphide bonded structures”

because even without a disulphide bond SurA appears to be positioned to interact with POTRA1. The authors should either remove this sentence or modify it to indicate that the results are consistent with other data (e.g., the crosslinking data) that indicate that SurA binds to BamA POTRA1.

We have changed our sentence to say ‘These release complexes support...’ (now line 344)

6) Fig. 7 and paragraph that starts at line 356: Because of the inherent limitations of the data, the authors should clearly state that the ‘Wait, Arrival, Handover and Release’ stages are a model, and not a sequence of events that has been clearly demonstrated by the results. The legend to Fig. 7 should also indicate that the authors are displaying a model.

We have amended the Figure 7 legend to say Model and amended line 362 to say we describe details for ‘putative OMP delivery.’

7) Regarding my original comment 16: I do not think that the authors have addressed this concern adequately. The new statements on lines 375-377 and 438-443 are very vague and confusing, and do not incorporate their work into the context of the whole field. It would be fair to say that we know more about the environment of the periplasm than that it is complex and crowded. As I originally noted, there is a lot of evidence in the literature that specific chaperones such as Skp and BepA play important roles in OMP biogenesis, that many OMPs can be assembled at least to some degree without SurA (and possibly without any other chaperone), and that the interaction of SurA with BamA is “timed” so that defective OMPs can be removed by other factors. A new review by Devlin and Fleming (TIBS 2024, doi: 10.1016/j.tibs.2024.03.015) documents some of the unique and redundant properties of periplasmic chaperones that have been discovered. I am not suggesting that the authors write a paragraph about other chaperones, but I think they should at least add a couple of sentences that clearly state that SurA is not essential for OMP assembly, that SurA might not bind to BamA and a partially folded OMP indefinitely, and that specific chaperones have distinct functions in OMP biogenesis.

We have amended the final paragraph to better place our work in the wider context of the field. (lines 445 to 450) and cited the review here too.

Minor comment:

1) Line 124: For clarity the authors should state that SurA residue 23 is the N-terminal residue of the mature protein because the first 22 residues are the cleaved signal peptide.

We have added this in as suggested (now line 122-124).